# PEAR: PRIMITIVE ENABLED ADAPTIVE RELABELING FOR BOOSTING HIERARCHICAL REINFORCEMENT LEARNING

## ABSTRACT

Hierarchical reinforcement learning (HRL) has the potential to solve complex long horizon tasks using temporal abstraction and increased exploration. However, hierarchical agents are difficult to train due to inherent non-stationarity. We present primitive enabled adaptive relabeling (PEAR), a two-phase approach where we first perform adaptive relabeling on a few expert demonstrations to generate efficient subgoal supervision, and then jointly optimize HRL agents by employing reinforcement learning (RL) and imitation learning (IL). We perform theoretical analysis to $(i)$ bound the sub-optimality of our approach, and $(ii)$ derive a generalized plug-and-play framework for joint optimization using RL and IL. PEAR uses a handful of expert demonstrations and makes minimal limiting assumptions on the task structure. Additionally, it can be easily integrated with typical model free RL algorithms to produce a practical HRL algorithm. We perform experiments on challenging robotic environments and show that PEAR is able to solve tasks that require long term decision making. We empirically show that PEAR exhibits improved performance and sample efficiency over previous hierarchical and non-hierarchical approaches. We also perform real world robotic experiments on complex tasks and demonstrate that PEAR consistently outperforms the baselines.

## 1 INTRODUCTION

Recently, reinforcement learning has been successfully applied to a number of short-horizon robotic manipulation tasks (Rajeswaran et al., 2017; Kalashnikov et al., 2018; Gu et al., 2016; Levine et al., 2015). However, long horizon tasks require long-term planning and are harder to solve (Gupta et al., 2019b) due to inherent issues like credit assignment and ineffective exploration. Consequently, such tasks require large number of environment interactions for learning, especially in sparse reward scenarios (Andrychowicz et al., 2017). Hierarchical reinforcement learning (HRL) (Sutton et al., 1999; Dayan & Hinton, 1993; Vezhnevets et al., 2017; Klissarov et al., 2017; Bacon et al., 2016) holds the promise of solving complex tasks by employing temporal abstraction and improved exploration (Nachum et al., 2019). In goal-conditioned feudal architecture (Dayan & Hinton, 1993; Vezhnevets et al., 2017), higher level policy predicts subgoals for the lower primitive, which in turn tries to achieve them by executing primitive actions directly on the environment. Unfortunately, HRL suffers from non-stationarity(Nachum et al., 2018; Levy et al., 2017) when multiple levels are trained simultaneously. Due to continuously changing policies, previously collected off-policy experience is rendered obsolete, leading to unstable higher level state transition and reward functions.

A particular class of hierarchical approaches (Gupta et al., 2019a; Fox et al., 2017; Krishnan et al., 2019) segment expert demonstrations into subgoal transition dataset, and consequently leverage the subgoal dataset to bootstrap learning. Ideally, the segmentation process should produce subgoals at appropriate level of difficulty for the lower primitive, in order to properly balance the task split between hierarchical levels. One possible approach of task segmentation is to perform fixed window based relabeling (Gupta et al., 2019a) on expert demonstrations. Despite being simple, this approach is effectively a brute force segmentation approach which may generate subgoals that are either too easy or too hard with respect to the current goal achieving ability of the continuously changing lower primitive, thus leading to degenerate solutions. This leads to the following question: can we do better than fixed relabeling and devise a HRL approach for efficient task segmentation?

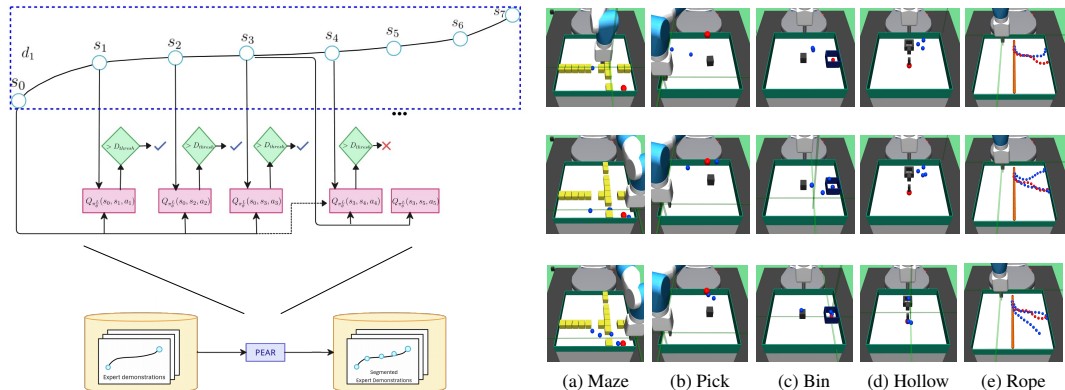

Figure 1: **Adaptive Relabeling Overview**: We segment expert demonstrations by consecutively passing demonstration states as subgoals (for $i = 1$ to $7$) to lower primitive, and finding the state where $Q_{\pi L}(s, s_i, a_i) < Q_{thresh}$ (here $s_i = s_4$). Since $s_3$ was the last reachable subgoal, it is selected as subgoal for initial state $s_0$. The transition is added to $D_g$, and $s_3$ is selected as the new initial state.

Figure 2: **Subgoal evolution**: With training, as lower primitive improves, higher level subgoal predictions (blue spheres) become better and harder, while always being achievable by lower primitive. Row 1 depicts initial training, Row 2 depicts mid-way through training, and Row 3 depicts end of training. This generates a curriculum of achievable subgoals for lower primitive (red spheres represent final goal).

As Greek philosopher Heraclitus said: *there is nothing permanent except change*. Hence, our idea considers the changing lower primitive and dynamically generates efficient subgoals in consonance with the current goal achieving capability of the lower primitive. In our approach, the action value function of lower primitive is used to perform *adaptive relabeling* on expert demonstrations to dynamically generate a curriculum of achievable subgoals for the lower primitive. This subgoal dataset is then used to train an imitation learning based regularizer, which is used to jointly optimize off-policy RL objective with IL regularization. Our approach thus combines HRL with primitive enabled imitation learning regularization to devise an elegant HRL algorithm that ameliorates non-stationarity. We call our approach: *primitive enabled adaptive relabeling (PEAR)* for boosting HRL.

Our major contributions are as follows: $(i)$ our adaptive relabeling based approach generates efficient higher level subgoal supervision considering the current goal achieving capability of lower primitive, $(ii)$ we propose a generalized plug-and-play framework for joint optimization using RL and IL, $(iii)$ we derive sub-optimality bounds to theoretically justify the benefits of periodic repopulation using adaptive relabeling, thus devising a practical HRL algorithm, $(iv)$ we perform extensive experimentation on complex robotic tasks: maze navigation, pick and place, bin, hollow, rope manipulation and franka kitchen to empirically demonstrate better performance and sample efficiency over prior hierarchical and non-hierarchical baselines on all tasks, and $(v)$ we perform real world experiments on robotic pick and place, bin and rope manipulation tasks in Section 5 to show that PEAR shows impressive generalization in complex real world scenarios. In summary, we propose a theoretically justified practical HRL algorithm for solving complex long horizon tasks.

## 2 RELATED WORK

Hierarchical reinforcement learning (HRL) framework (Barto & Mahadevan, 2003; Sutton et al., 1999; Parr & Russell, 1998; Dietterich, 1999) promises the advantages of temporal abstraction and increased exploration (Nachum et al., 2019). The options architecture (Sutton et al., 1999; Bacon et al., 2016; Harutyunyan et al., 2017; Harb et al., 2017; Harutyunyan et al., 2019; Klissarov et al., 2017) learns temporally extended macro actions and termination function to propose an elegant hierarchical framework. However, such approaches may produce degenerate solutions in the absence of proper regularization. Some typical approaches restrict the problem search space by greedily solving for specific goals (Kaelbling, 1993; Foster & Dayan, 2002), which has also been extended to hierarchical RL (Wulfmeier et al., 2019; 2020; Ding et al., 2019). In goal-conditioned hierarchical feudal learning (Dayan & Hinton, 1993; Vezhnevets et al., 2017), the higher level agent produces subgoals for the lower primitive, which in turn executes atomic actions on the environment. However, off-policy feudal HRL approaches are cursed by non-stationarity issue. Some prior approaches (Nachum et al., 2018; Levy et al., 2017) deal with the non-stationarity by relabeling previously collected transitions for training goal-conditioned policies. In contrast, our proposed

approach deals with non-stationarity by leveraging adaptive relabeling for periodically producing achievable subgoals, and subsequently using an imitation learning based regularizer in our joint optimization based approach. We empirically show in section 5 that our regularization based approach outperforms relabeling based hierarchical approaches on a number of complex long horizon tasks.

Prior methods (Rajeswaran et al., 2017; Nair et al., 2017; Hester et al., 2017) leverage expert demonstrations to improve sample efficiency and accelerate learning. Prior work uses imitation learning to bootstrap learning (Shiarlis et al., 2018; Krishnan et al., 2017; 2019; Kipf et al., 2019). Other approaches use fixed relabeling (Gupta et al., 2019a) for performing task segmentation. However, such approaches may cause unbalanced task split between hierarchical levels. In contrast, our approach sidesteps this limitation by segmenting expert demonstration trajectories according to current lower primitive. Intuitively, this enables balanced task split, thereby avoiding degenerate solutions. Recent approaches restrict subgoal space using adjacency constraints (Zhang et al., 2020), employ graph based approaches for decoupling task horizon (Lee et al., 2022), or incorporate imagined subgoals combined with KL-constrained policy iteration scheme (Chane-Sane et al., 2021). However, such approaches assume additional environment constraints and only work on relatively shorter horizon tasks with limited complexity. (Kreidieh et al., 2019) is an inter-level cooperation based approach for generating achievable subgoals, However, the approach requires extensive exploration for selecting good subgoals, whereas our approach rapidly enables effective subgoal generation using primitive enabled adaptive relabeling. In order to accelerate RL, recent work firstly learns behavior skill priors (Pertsch et al., 2020; Singh et al., 2020) from expert data or pre-trains policies over a related task, and then later fine-tunes using RL. Such approaches largely depend on policies learnt during pre-training, and are hard to train when the source and target task distributions are dis-similar. Some hierarchical approaches hand-design action primitives (Dalal et al., 2021; Nasiriany et al., 2021), and then predict arguments for selecting from among the primitives. While this makes the task easier for higher level policy, explicitly designing action primitives can be tedious for hard tasks, or lead to sub-optimal policies. Since PEAR learns multi-level policies in parallel, the lower level policies can learn the required optimal behavior, thus avoiding the issues inherent with previous approaches.

## 3 BACKGROUND

**Off-policy Reinforcement Learning** We define our goal-conditioned off-policy RL setup as follows: *Universal Markov Decision Process* (UMDP) (Schaul et al., 2015) are markov decision processes augmented with the goal space $G$, where $M = (S, A, P, R, \gamma, G)$. Here, $S$ is state space, $A$ is action space, $P(s'|s,a)$ is the state transition probability function, $R$ is reward function, and $\gamma$ is discount factor. $\pi(a|s,g)$ represents the goal-conditioned policy which predicts the probability of taking action $a$ when the state is $s$ and goal is $g$. The overall objective is to maximize expected future discounted reward distribution: $J = (1-\gamma)^{-1}\mathbb{E}_{s\sim d^\pi, a\sim\pi(a|s,g), g\sim G}[r(s_t, a_t, g)]$.

**Hierarchical Reinforcement Learning** In our goal-conditioned HRL setup, the overall policy $\pi$ is divided into multi-level policies. We consider bi-level scheme, where the higher level policy $\pi^H(s_g|s,g)$ predicts subgoals $s_g$ for the lower primitive, and lower primitive $\pi^L(a|s,s_g)$ executes primitive actions $a$ on the environment. $\pi^H$ generates subgoals $s_g$ after every $c$ timesteps and $\pi^L$ tries to achieve $s_g$ within $c$ timesteps. $\pi^H$ gets sparse extrinsic reward $r_{ex}$ from the environment, whereas $\pi^L$ gets sparse intrinsic reward $r_{in}$ from $\pi^H$. $\pi^L$ gets rewarded with reward 0 if the agent reaches within $\delta^L$ distance of the predicted subgoal $s_g$, and $-1$ otherwise: $r_{in} = -1(\|s_t - s_g\|_2 > \delta^L)$. Similarly, $\pi^H$ gets extrinsic reward 0 if the achieved goal is within $\delta^H$ distance of the final goal $g$, and $-1$ otherwise: $r_{ex} = -1(\|s_t - g\|_2 > \delta^H)$. We assume access to a small number of directed expert demonstrations states $D = \{e^i\}_{i=1}^N$, where $e^i = (s_0^e, s_1^e, \ldots, s_{T-1}^e)$.

## 4 METHODOLOGY

Here, we explain our proposed primitive enabled adaptive relabeling *PEAR* approach, which leverages a handful of expert demonstrations $D$ to solve long horizon tasks. We propose a two step approach: $(i)$ the current lower primitive $\pi^L$ is used to adaptively relabel expert demonstrations to generate efficient subgoal supervision $D_g$, and $(ii)$ typical reinforcement learning objective is jointly optimized with additional imitation learning based regularization objective using $D_g$. We perform theoretical analysis to $(i)$ bound the sub-optimality of our approach, and $(ii)$ propose a

practical plug-and-play based framework for joint optimization using RL and IL, where we can plug in typical off-policy RL and IL algorithms to generate novel joint optimization based algorithms.

---

**Algorithm 1** Adaptive Relabeling

1: Initialize $D_g = \{\}$
2: **for** each $e = (s_0^e, s_1^e, \ldots, s_{T-1}^e)$ in $\mathcal{D}$ **do**
3:     Initial state index $init \leftarrow 0$
4:     Subgoal transitions $D_g^e = \{\}$
5:     **for** i = 1 **to** $T-1$ **do**
6:         # Find $Q_{\pi^L}$ values for demo subgoals
7:         Compute $Q_{\pi^L}(s_{init}^e, s_i^e, a_i)$
8:             where $a_i = \pi^L(s_{i-1}^e, s_i^e)$
9:         # Find first subgoal s.t. $Q_{\pi^L} < Q_{th}$
10:        **if** $Q_{\pi^L}(s_{init}^e, s_i^e, a_i) < Q_{th}$ **then**
11:           **for** j = $init$ **to** $i-1$ **do**
12:              **for** k = $(init+1)$ **to** $i-1$ **do**
13:                 # Add the transition to $D_g^e$
14:                 Add $(s_j, s_{i-1}, s_k)$ to $D_g^e$
15:         Initial state index $init \leftarrow (i-1)$
16:     # Add selected transitions to $D_g$
17:     $D_g \leftarrow D_g \cup D_g^e$

---

**Algorithm 2** PEAR

1: Initialize $D_g = \{\}$
2: **for** $i = 1 \ldots N$ **do**
3:     **if** $i\%p == 0$ **then**
4:         Clear $D_g$
5:         Populate $D_g$ via adaptive relabeling
6:     Collect experience using $\pi^H$ and $\pi^L$
7:     Update lower primitive via SAC and IL
8:         regularizer using $D$ (Eq 6 or Eq 8)
9:     Sample transitions from $D_g$
10:    Update higher policy via SAC and IL
11:        regularizer using $D_g$ (Eq 5 or Eq 7)

---

### 4.1 PRIMITIVE ENABLED ADAPTIVE RELABELING

PEAR uses the lower primitive's action value function $Q_{\pi^L}(s, s_i^e, a_i)$ to parse the expert demonstration trajectories $D$ and generate efficient subgoal transition dataset $D_g$. In a typical goal-conditioned RL setting, $Q_{\pi^L}(s, s_i^e, a_i)$ describes the expected cumulative reward when the input starting state and subgoal are $s$ and $s_i^e$, and the lower primitive takes action $a_i$ while following policy $\pi^L$ in the episode. The expert demonstrations states $s_i^e$ are passed as subgoals, and $Q_{\pi^L}(s, s_i^e, a_i)$ computes the expected cumulative reward when start state is $s$, subgoal is $s_i^e$ and the next primitive action is $a_i$. Intuitively, a high value of $Q_{\pi^L}(s, s_i^e, a_i)$ implies that the current lower primitive considers $s_i^e$ to be a good (highly rewarding and achievable) subgoal from current state $s$, since it expects to achieve a high intrinsic reward for this subgoal from the higher policy. Conversely, a low value of $Q_{\pi^L}(s, s_i^e, a_i)$ implies that the lower primitive considers $s_i^e$ to be a bad (low rewarding or unachievable) subgoal, since it expects to achieve a low intrinsic reward for $s_i^e$ from current state $s$. Hence, $Q_{\pi^L}(s, s_i^e, a_i)$ considers goal achieving capability of current lower primitive for populating $D_g$. We depict a single pass of adaptive relabeling in Figure 1 and explain the procedure in detail below.

Consider the expert demonstration dataset $D = \{e^j\}_{i=1}^N$, where each trajectory $e^j = (s_0^e, s_1^e, \ldots, s_{T-1}^e)$. Let the initial state be $s_0^e$. In the adaptive relabeling procedure, we incrementally provide demonstration states $s_i^e$ for $i = 1$ to $T-1$ as subgoals to lower primitive's action value function $Q_{\pi^L}(s_0^e, s_i^e, a_i)$, where $a_i = \pi^L(s = s_{i-1}^e, g = s_i^e)$. At every step, we compare $Q_{\pi^L}(s_0^e, s_i^e, a_i)$ to the environment specific $Q_{thresh}$ value. If $Q_{\pi^L}(s_0^e, s_i^e, a_i) >= Q_{thresh}$, we move on to next expert demonstration state $s_{i+1}^e$. Otherwise if $Q_{\pi^L}(s_0^e, s_i^e, a_i) < Q_{thresh}$, we consider $s_{i-1}^e$ as a good subgoal for initial state (since it was the last subgoal with $Q_{\pi^L}(s_0^e, s_{i-1}^e, a_i) >= Q_{thresh}$), and use it to compute subgoal transitions for populating $D_g$. Subsequently, we repeat the same procedure with $s_{i-1}^e$ as the new initial state, until the episode terminates. This is also depicted in Algorithm 1.

HRL approaches suffer from non-stationarity due to unstable higher level station transition and reward functions. In off-policy RL, this occurs as the previously collected experience is rendered obsolete due to continuously changing lower primitive. Similarly, the subgoal transitions in $D_g$ collected using adaptive relabeling also become outdated with changing lower primitive and $Q_{\pi^L}(s_0^e, s_i^e, a_i)$. We propose to mitigate this non-stationarity by periodically re-populating subgoal transition dataset $D_g$ after every $p$ timesteps according to the goal achieving capability of the current lower primitive. Since the lower primitive continuously improves with training and gets better at achieving harder subgoals, $Q_{\pi^L}$ always picks reachable subgoals of appropriate difficulty, according to the current goal reaching ability of the lower primitive. This generates a natural curriculum of subgoals for

lower primitive. Intuitively, $D_g$ always contains achievable subgoals for the current lower primitive, thereby mitigating the non-stationarity issue. The pseudocode for PEAR is given in Algorithm 2. Figure 2 shows the qualitative evolution of subgoals during training in our experiments.

Our adaptive relabeling procedure uses $Q_{\pi^L}(s_0^e, s_i^e, a_i)$ to select efficient subgoals when the expert state $s_i^e$ is within the training distribution of states used to train the lower primitive. However, if the expert states are outside the training distribution, $Q_{\pi_L}$ might erroneously over-estimate the values on out-of-distribution states, which might result in poor subgoal selection. In order to address this over-estimation issue, we employ an additional margin classification objective(Piot et al., 2014), where along with the standard $Q_{SAC}$ objective, we also use an additional margin classification objective to yield the following optimization objective $\bar{Q}_{\pi^L} = Q_{SAC} +$

$$\arg\min_{Q_{\pi_L}} \max_{\pi^L}(\mathbb{E}_{(s_0^e, \cdot, \cdot) \sim D_g, s_i^e \sim \pi^H, a_i \sim \pi^L}[Q_{\pi^L}(s_0^e, s_i^e, a_i)] - \mathbb{E}_{(s_0^e, s_i^e, \cdot) \sim D_g, a_i \sim \pi^L}[Q_{\pi^L}(s_0^e, s_i^e, a_i)])$$

This surrogate objective prevents over-estimation of $\bar{Q}_{\pi^L}$ by penalizing states that are out of the expert state distribution. We found this objective to improve performance and stabilize learning. Next, we explain the details and rationale behind our joint optimization objective.

## 4.2 JOINT OPTIMIZATION

In this section, we explain our joint optimization objective comprising RL objective with IL based regularization. We consider both behavior cloning (BC) and inverse reinforcement learning (IRL) regularization. Henceforth, PEAR-IRL will represent PEAR with IRL regularization and PEAR-BC will represent PEAR with BC regularization. We first explain BC regularization objective, and then explain IRL regularization objectives for both hierarchical levels.

For the BC objective, let $(s^e, s_g^e, s_{next}^e) \sim D_g$ represent a higher level subgoal transition from $D_g$ where $s^e$ is current state, $s_{next}^e$ is next state, $g^e$ is final goal and $s_g^e$ is subgoal supervision. Let $s_g$ be the subgoal predicted by the high level policy $\pi_{\theta_H}^H(\cdot|s^e, g^e)$ with parameters $\theta_H$. The BC regularization objective for higher level is as follows:

$$\min_{\theta_H} J_{BC}^H(\theta_H) = \min_{\theta_H} \mathbb{E}_{(s^e, s_g^e, s_{next}^e) \sim D_g, s_g \sim \pi_{\theta_H}^H(\cdot|s^e, g^e)} ||s_g^e - s_g||^2 \tag{1}$$

Similarly, let $(s^f, a^f, s_{next}^f) \sim D_g^L$ represent lower level expert transition where $s^f$ is current state, $s_{next}^f$ is next state, $g^f$ is goal and $a$ is the primitive action predicted by $\pi_{\theta_L}^L(\cdot|s^f, s_g^e)$ with parameters $\theta_L$. The lower level BC regularization objective is as follows:

$$\min_{\theta_L} J_{BC}^L(\theta_L) = \min_{\theta_L} \mathbb{E}_{(s^f, a^f, s_{next}^f) \sim D_g^L, a \sim \pi_{\theta_L}^L(\cdot|s^f, s_g^e)} ||a^f - a||^2 \tag{2}$$

We now consider the IRL objective, which is implemented as a GAIL (Ho & Ermon, 2016) objective implemented using LSGAN (Mao et al., 2016). Let $\mathbb{D}_\epsilon^H$ be the higher level discriminator with parameters $\epsilon_H$. Let $J_D^H$ represent higher level IRL objective, which depends on parameters $(\theta_H, \epsilon_H)$. The higher level IRL regularization objective is as follows:

$$\max_{\theta_H} \min_{\epsilon_H} J_D^H(\theta_H, \epsilon_H) = \max_{\theta_H} \min_{\epsilon_H} \frac{1}{2} \mathbb{E}_{(s^e, \cdot, \cdot) \sim D_g, s_g \sim \pi_{\theta_H}(\cdot|s^e, g^e)}[\mathbb{D}_{\epsilon_H}^H(\pi_{\theta_H}^H(\cdot|s^e, g^e)) - 0]^2$$
$$+ \frac{1}{2} \mathbb{E}_{(s^e, s_g^e, \cdot) \sim D_g}[\mathbb{D}_{\epsilon_H}^H(s_g^e) - 1]^2 \tag{3}$$

Similarly, for lower level primitive, let $\mathbb{D}_{\epsilon_L}^L$ be the lower level discriminator with parameters $\epsilon_L$. Let $J_D^L$ represent lower level IRL objective, which depends on parameters $(\theta_L, \epsilon_L)$. The lower level IRL regularization objective is as follows:

$$\max_{\theta_L} \min_{\epsilon_L} J_D^L(\theta_L, \epsilon_L) = \max_{\theta_L} \min_{\epsilon_L} \frac{1}{2} \mathbb{E}_{(s^f, \cdot, \cdot) \sim D_g^L, a \sim \pi_{\theta_L}^L(\cdot|s^f, s_g^e)}[\mathbb{D}_{\epsilon_L}^L(\pi_{\theta_L}^L(\cdot|s^f, s_g^e)) - 0]^2$$
$$+ \frac{1}{2} \mathbb{E}_{(s^f, a^f, \cdot) \sim D_g^L}[\mathbb{D}_{\epsilon_L}^L(a^f) - 1]^2 \tag{4}$$

Finally, we describe our joint optimization objective for hierarchical policies. Let the off-policy RL objective be $J_{\theta_H}^H$ and $J_{\theta_L}^L$ for higher and lower policies. The joint optimization objectives using BC regularization for higher and lower policies are provided in Equations 5 and 6.

$$\max_{\theta_H}(J_{\theta_H}^H - \psi * J_{BC}^H(\theta_H)) \tag{5}$$

$$\max_{\theta_L}(J_{\theta_L}^L - \psi * J_{BC}^L(\theta_L)) \tag{6}$$

The joint optimization objectives using IRL regularization for higher and lower policies are provided in Equations 7 and 8.

$$\min_{\epsilon_H}\max_{\theta_H}(J_{\theta_H}^H + \psi * J_D^H(\theta_H, \epsilon_H)) \tag{7}$$

$$\min_{\epsilon_L}\max_{\theta_L}(J_{\theta_L}^L + \psi * J_D^L(\theta_L, \epsilon_L)) \tag{8}$$

Here, $\psi$ is regularization weight hyper-parameter. We perform experiments to choose $\psi$ in Section 5.

### 4.3 SUBOPTIMALITY ANALYSIS AND PLUG-AND-PLAY FRAMEWORK FOR JOINT OPTIMIZATION

In this section, we perform theoretical analysis to $(i)$ derive sub-optimality bounds for our proposed joint optimization objective and show how our periodic re-population based approach affects performance, and $(ii)$ propose a generalized plug-and-play framework for joint optimization using RL and IL. Let $\pi^*$ and $\pi^{**}$ be unknown higher level and lower level optimal policies. Let $\pi_{\theta_H}^H$ be our high level policy and $\pi_{\theta_L}^L$ be our lower primitive policy, where $\theta_H$ and $\theta_L$ are trainable parameters. $D_{TV}(\pi_1, \pi_2)$ denotes total variation divergence between probability distributions $\pi_1$ and $\pi_2$. Let $\kappa$ be an unknown distribution over states and actions, $G$ be goal space, $s$ be current state, and $g$ the final episodic goal. We will use $\kappa$ in an importance sampling ratio later to avoid sampling from the unknown optimal policy. The higher level policy predicts subgoals $s_g$ for the lower primitive which executes for $c$ timesteps to yield sub-trajectories $\tau$. Let $\Pi_D^H$ and $\Pi_D^L$ be some unknown higher and lower level probability distributions over policies from which we can sample policies $\pi_D^H$ and $\pi_D^L$. Let us assume that policies $\pi_D^H$ and $\pi_D^L$ represent the policies from higher and lower level datasets $D_H$ and $D_L$ respectively. Although $D_H$ and $D_L$ may represent any datasets, in our discussion, we use them to represent higher and lower level expert demonstration datasets. Firstly, we extend the $\phi_D$-common definition from (Ajay et al., 2020) to goal-conditioned policies:

**Definition 1.** $\pi^*$ is $\phi_D$-common in $\Pi_D^H$, if $\mathbb{E}_{s\sim\kappa, \pi_D^H\sim\Pi_D^H, g\sim G}[D_{TV}(\pi^*(\tau|s,g)||\pi_D^H(\tau|s,g))] \le \phi_D$

Now, we define the suboptimality of policy $\pi$ with respect to optimal policy $\pi^*$ as:

$$Subopt(\theta) = |J(\pi^*) - J(\pi)| \tag{9}$$

**Theorem 1.** Assuming optimal policy $\pi^*$ is $\phi_D$ common in $\Pi_D^H$, the suboptimality of higher policy $\pi_{\theta_H}^H$, over $c$ length sub-trajectories $\tau$ sampled from $d_c^{\pi^*}$ can be bounded as:

$$|J(\pi^*) - J(\pi_{\theta_H}^H)| \le \lambda_H * \phi_D + \lambda_H * \mathbb{E}_{s\sim\kappa, \pi_D^H\sim\Pi_D^H, g\sim G}[D_{TV}(\pi_D^H(\tau|s,g)||\pi_{\theta_H}^H(\tau|s,g))] \tag{10}$$

where $\lambda_H = \frac{2}{(1-\gamma)(1-\gamma^c)}R_{max}\|\frac{d_c^{\pi^*}}{\kappa}\|_\infty$

Similarly, the suboptimality of lower primitive $\pi_{\theta_L}^L$ can be bounded as:

$$|J(\pi^{**}) - J(\pi_{\theta_L}^L)| \le \lambda_L * \phi_D + \lambda_L * \mathbb{E}_{s\sim\kappa, \pi_D^L\sim\Pi_D^L, s_g\sim\pi_{\theta_H}^H}[D_{TV}(\pi_D^L(\tau|s,s_g)||\pi_{\theta_L}^L(\tau|s,s_g))] \tag{11}$$

where $\lambda_L = \frac{2}{(1-\gamma)^2}R_{max}\|\frac{d_c^{\pi^{**}}}{\kappa}\|_\infty$

The proofs for Equations 10 and 11 are provided in Appendix A.1. In Equation 10, the suboptimality of $\pi_{\theta_H}^H$ is bounded by the two terms on RHS, which we discuss in detail below.

We firstly focus on the first term in which is dependent on $\phi_D$. In our discussion $D$ is replaced by dataset $D_g$ populated using the current lower primitive, hence $\phi_D$ becomes $\phi_{D_g}$. In Theorem 1, we assume the optimal policy $\pi^*$ to be $\phi_{D_g}$ common in $\Pi_D^H$. Since $\phi_{D_g}$ denotes the upper bound on the expected TV divergence between $\pi^*$ and $\pi_D^H$, $\phi_{D_g}$ provides a quality measure of the subgoal dataset $D_g$ populated using adaptive relabeling. A lower value of $\phi_{D_g}$ implies that the optimal policy $\pi^*$ is closely represented by $D_g$, or in other words, the samples from $D_g$ are near optimal. Intuitively, since the lower primitive improves with training and is able to achieve harder subgoals, and since

$D_g$ is re-populated using the improved lower primitive after every $p$ timesteps, $\pi_{D_g}$ continually gets closer to $\pi^*$, resulting in decrease in value of $\phi_D$. This implies that the suboptimality bound in Equation 10 gets tighter, and consequently $J(\pi_{\theta_H}^H)$ gets closer to optimal $J(\pi^*)$ objective. Hence, our periodic re-population based approach generates a natural curriculum of achievable subgoals for the lower primitive, which continuously improves the performance by tightening the upper bound.

Now, we focus on the second term in Equation 10, which is TV divergence between $\pi_D^H(\tau|s,g)$ and $\pi_{\theta_H}^H(\tau|s,g)$ with expectation over $s \sim \kappa, \pi_D^H \sim \Pi_D^H, g \sim G$. As before, $D$ is replaced by dataset $D_g$. This term can be viewed as imitation learning (IL) objective between expert demonstration policy $\pi_{D_g}^H$ and current policy $\pi_{\theta_H}^H$, where TV divergence is the distance measure. Due to this IL regularization objective, as policy $\pi_{\theta_H}^H$ gets closer to expert distribution policy $\pi_{D_g}^H$ with training, the LHS sub-optimality bounds get tighter. Thus, our proposed periodic re-population and IL regularization tighten the sub-optimality bounds in Equation 10 with training, thus improving performance.

We now derive our generalized plug-and-play framework for the joint optimization objective, where we can plug in off the shelf RL and IL methods to yield a generally applicable practical HRL algorithm. Considering the idea that sub-optimality is positive, we can derive the following equation:

$$J(\pi^*) \geq J(\pi_{\theta_H}^H) - \lambda_H * \phi_D - \lambda_H * \mathbb{E}_{s \sim \kappa, \pi_D^H \sim \Pi_D^H, g \sim G}[d(\pi_D^H(\tau|s,g)||\pi_{\theta_H}^H(\tau|s,g))] \quad (12)$$

where (considering $\pi_D^H(\tau|s,g)$ as $\pi_A$ and $\pi_{\theta_H}^H(\tau|s,g)$) as $\pi_B$, $d(\pi_A||\pi_B) = D_{TV}(\pi_A||\pi_B)$

Notably, the second term $\lambda_H * \phi_D$ in RHS of Equation 12 is constant for a given dataset $D_g$. Equation 12 can be perceived as a minorize maximize algorithm which intuitively means: the overall objective can be optimized by $(i)$ maximizing the objective $J(\pi_{\theta_H}^H)$ via RL, and $(ii)$ minimizing the distance measure $d$ between $\pi_D^H$ and $\pi_{\theta_H}^H$. This formulation serves as a plug-and-play framework where we can plug in RL algorithm of choice for our off-policy RL objective $J(\pi_{\theta_H}^H)$, and distance function of choice for distance measure $d$ to yield various joint optimization objectives.

In our setup, we plug in entropy regularized Soft Actor Critic (Haarnoja et al., 2018a) to maximize $J(\pi_{\theta_H}^H)$. Notably, different parameterizations of $d$ yield different imitation learning regularizers. When $d$ is formulated as Kullback–Leibler divergence, the IL regularizer takes the form of behavior cloning (BC) objective (Nair et al., 2017) (which results in PEAR-BC), and when $d$ is formulated as Jensen-Shannon divergence, the imitation learning objective takes the form of inverse reinforcement learning (IRL) objective (which results in PEAR-IRL). We consider both these objectives in Section 5 and explicitly provide empirical performance results.

## 5 EXPERIMENTS

In this section, we empirically answer the following questions: $(i)$ does adaptive relabeling approach outperform fixed relabeling based approaches? $(ii)$ is PEAR able to mitigate non-stationarity? and $(iii)$ does IL regularization boost performance in solving complex long horizon tasks. We accordingly perform experiments on six Mujoco (Todorov et al., 2012) environments: $(i)$ maze navigation, $(ii)$ pick and place, $(iii)$ bin, $(iv)$ hollow, $(v)$ rope manipulation, and $(vi)$ franka kitchen, and demonstrate that our approach consistently outperforms the baselines.

**Environment details:** We provide extensive environment and implementation details in the Appendix A.3, where we provide the details of all the tasks and final goal configurations. The maximum task horizon $T$ is kept at 225, 50, 60, 100 25, and 280 timesteps, and the lower primitive is allowed to execute for $c$ timesteps, ie 15, 7, 6, 10, 5,and 17 for the maze, pick and place, bin, hollow, rope and kitchen respectively. We use 28 expert demos for franks kitchen task and 100 demos in all other tasks, and provide the procedures for collecting expert demos for all tasks in Appendix A.2.

**Implementation details:** In our experiments, we use off-policy Soft Actor Critic (Haarnoja et al., 2018b) for optimizing RL objective, using Adam (Kingma & Ba, 2014) optimizer. The actor, critic and discriminator networks are formulated as 3 layer fully connected neural networks with 512 neurons in each layer. When calculating $p$, we normalize $Q_{\pi^L}$ values of a trajectory before comparing with $Q_{thresh}$: $((Q_{\pi^L}(s_0^e, s_i^e, a_i) - min\_value)/max\_value) * 100$ for $i = 1$ to $T - 1$.

**Evaluation and results:** In Table 1, we report the success rate performance of our method and other baselines averaged over 5 seeds, and evaluated over $N = 100$ random episodic rollouts. Firstly, we

Table 1: Success rate comparison

|  | Maze | Pick Place | Bin | Hollow | Rope | Kitchen |
|---|---|---|---|---|---|---|
| PEAR-IRL | **0.84 ± 0.04** | **0.92 ± 0.02** | **0.79 ± 0.05** | **0.78 ± 0.27** | **0.33 ± 0.04** | **0.89 ± 0.06** |
| PEAR-BC | 0.67 ± 0.07 | **0.48 ± 0.3** | **0.38 ± 0.19** | **0.33 ± 0.03** | 0.32 ± 0.04 | **1.0 ± 0.0** |
| RPL | 0.58 ± 0.09 | 0.28 ± 0.17 | 0.0 ± 0.0 | 0.0 ± 0.0 | 0.13 ± 0.07 | 0.08 ± 0.1 |
| HAC | 0.6 ± 0.23 | 0.0 ± 0.0 | 0.0 ± 0.0 | 0.1 ± 0.0 | 0.02 ± 0.01 | 0.0 ± 0.0 |
| RAPS | **0.81 ± 0.06** | 0.0 ± 0.0 | 0.0 ± 0.0 | 0.0 ± 0.0 | - | 0.0 ± 0.0 |
| HIER-NEG | 0.01 ± 0.0 | 0.0 ± 0.0 | 0.0 ± 0.0 | 0.0 ± 0.0 | 0.01 ± 0.0 | 0.0 ± 0.0 |
| HIER | 0.02 ± 0.02 | 0.0 ± 0.0 | 0.0 ± 0.0 | 0.0 ± 0.0 | 0.01 ± 0.0 | 0.0 ± 0.0 |
| DAC | 0.02 ± 0.02 | 0.21 ± 0.06 | 0.14 ± 0.09 | 0.0 ± 0.0 | 0.03 ± 0.01 | 0.0 ± 0.0 |
| FLAT | 0.01 ± 0.01 | 0.0 ± 0.0 | 0.0 ± 0.0 | 0.0 ± 0.0 | 0.03 ± 0.01 | 0.0 ± 0.0 |
| BC | 0.0 | 0.0 | 0.0 | 0.0 | 0.15 | 0.0 |

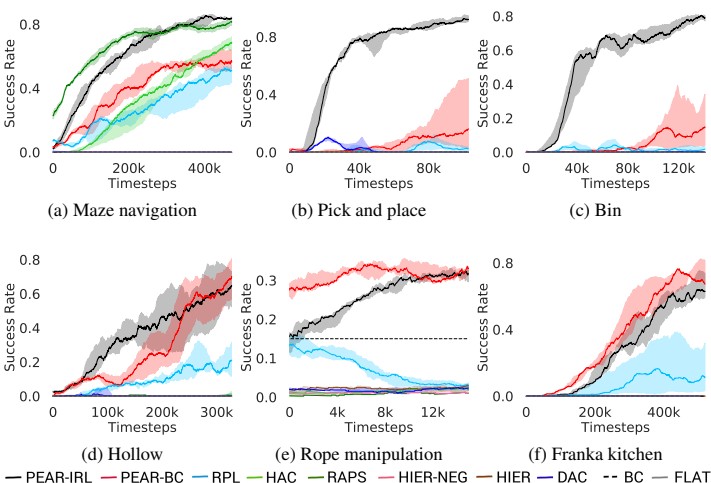

Figure 3: **Success rate comparison** in various environments vs number of timesteps over 5 seeds.

compare our method with Relay Policy Learning (RPL) to demonstrate that adaptive relabeling outperforms fixed relabeling. RPL (Gupta et al., 2019a) uses supervised pre-training followed by relay fine tuning. In order to ascertain fair comparisons, we use an ablation of RPL by removing supervised pre-training. Hierarchical actor critic (HAC) (Levy et al., 2017) deals with non-stationarity by relabeling transitions assuming an optimal lower primitive. We empirically found PEAR to consistently outperform HAC on all tasks, which shows that adaptive relabeling and IL regularization mitigate non-stationarity. We also consider RAPS (Dalal et al., 2021) baseline, which uses hand designed action primitives at the lower level. We do not evaluate RAPS in rope environment since hand designing action primitives is hard. The performance of RAPS depends on the quality of action primitives. We found that except maze navigation, PEAR significantly outperforms RAPS. PEAR outperforms hierarchical (HIER) baseline, and HIER-NEG baseline, which is a hierarchical baseline where the upper level is negatively rewarded if the lower primitive fails to achieve the subgoal. This demonstrates the importance of efficient subgoals supervision and subsequent IL regularization. We perform comparisons with Discriminator Actor Critic (DAC) (Kostrikov et al., 2018), which is a flat (single level) approach that leverages expert demos using a learned discriminator. We also compute a FLAT baseline that does not use expert demos. Our approach outperforms both these single level baselines by a significant margin, demonstrating the efficacy of our hierarchical approach with IL regularization. Finally, we also include a BC baseline and compare success rate performance. The training plots for the six environments are provided in Figure 3. In all experiments, PEAR exhibits faster convergence and consistently outperforms the baselines.

**Real world experiments:** We perform experiments on real world robotic pick and place, bin and rope environments (Fig 11). PEAR-IRL achieves accuracy of 0.8, 0.6, and 0.3, whereas PEAR-BC

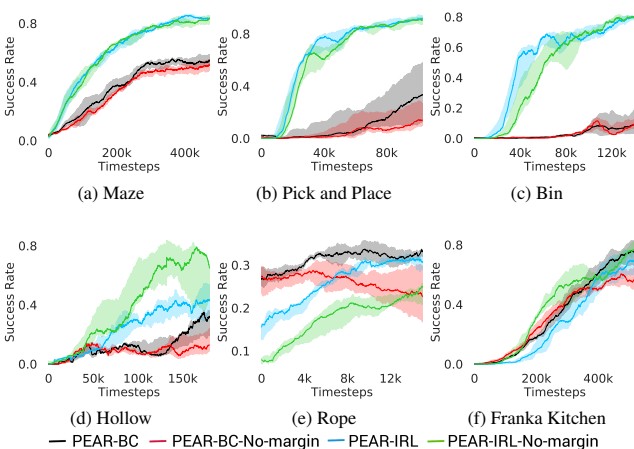

Figure 4: **Ablation experiments** in various environments: $(i)$ comparison between PEAR-IRL, PEAR-BC, PEAR-IRL-No-Margin and PEAR-BC-No-Margin with margin surrogate objective, $(ii)$ Row 2: $D_{thresh}$ hyper-parameter, and $(iii)$ Row 3: $p$ hyperparameter.

achieves accuracy of $0.8$, $0$, $0.3$ on pick and place, bin and rope environments. We also evaluate the next best performing RPL baseline, but it fails to achieve success in any of the tasks.

**Ablative analysis:** In order to analyse various design choices, first we compare PEAR-IRL and PEAR-BC (with margin classification objectives), with PEAR-IRL-No-Margin and PEAR-BC-No-Margin (without margin objectives) in Figure 4. PEAR-IRL and PEAR-BC almost always outperform PEAR-IRL-No-Margin and PEAR-BC-No-Margin, which shows that this objective is crucial for stable learning. We also analyse how varying $Q_{thresh}$ affects performance in Appendix A.4 Figure 9, and empirically find that even a low value of $0$ is sufficient for selecting good subgoals. Furthermore, when analyzing $p$ hyperparameter, we found that large values of $p$ are unable to generate good curriculum of subgoals (Appendix A.4 Figure 10), whereas small values of $p$ lead to frequent subgoal dataset re-population, impeding stable learning. We empirically choose optimal window size hyperparameter $k$ for RPL in Appendix A.4 Figure 5. We also evaluate optimal learning rate $\psi$ in Appendix A.4 Figure 6. If $\psi$ is too small, PEAR is unable to utilize IL regularization, whereas conversely if $\psi$ is too large, the learned policy might overfit. In order to verify the importance of adaptive relabeling, we replace it in PEAR-IRL by fixed window relabeling as in RPL (Gupta et al., 2019b), and call it PEAR-RPL. As shown in Appendix A.4 Figure 7, PEAR-IRL and PEAR-BC consistently outperform PEAR-RPL on all tasks. Furthermore, we perform ablations to deduce the optimal number of expert demos required for each task in Appendix A.4 Figure 8. We also provide qualitative visualizations in simulation in Appendix A.5.

# 6 DISCUSSION

**Limitations** In this work, we assume availability of directed expert demonstrations. While we do not consider undirected demonstrations in this work, we plan to explore this avenue in future. In our approach, $D_g$ is periodically re-populated, which is an additional overhead and might be a bottleneck in tasks where relabeling cost is high. Notably, in our setup, adaptive relabeling causes negligible overhead, as we pass the whole expert trajectory as a mini-batch for a single forward pass through lower primitive. Nevertheless, we plan to devise solutions to resolve this issue in future work.

**Conclusion and future work** We propose primitive enabled adaptive relabeling (PEAR), a HRL and IL based approach that performs adaptive relabeling on a handful of expert demonstrations to solve complex long horizon tasks. We perform comparisons with a various basselines and demonstrate that PEAR shows strong results in simulation and real world robotic tasks. In future work, we plan to address even harder sequential decision making tasks, and plan to analyse generalization beyond expert demonstrations. We hope that PEAR encourages future research in the area of adaptive relabeling and leads to efficient approaches for solving long horizon tasks.

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

# A APPENDIX

## A.1 SUB-OPTIMALITY ANALYSIS

Here, we present the proofs for Theorem 1 for higher and lower level policies, which provide sub-optimality bounds on the optimization objectives.

### A.1.1 SUB-OPTIMALITY PROOF FOR HIGHER LEVEL POLICY

The sub-optimality of upper policy $\pi_{\theta_H}^H$, over $c$ length sub-trajectories $\tau$ sampled from $d_c^{\pi^*}$ can be bounded as:

$$|J(\pi^*) - J(\pi_{\theta_H}^H)| \leq \lambda_H * \phi_D + \lambda_H * \mathbb{E}_{s \sim \kappa, \pi_D^H \sim \Pi_D^H, g \sim G}[D_{TV}(\pi_D^H(\tau|s,g)||\pi_{\theta_H}^H(\tau|s,g))]] \quad (13)$$

where $\lambda_H = \frac{2}{(1-\gamma)(1-\gamma^c)} R_{max} \|\frac{d_c^{\pi^*}}{\kappa}\|_\infty$

*Proof.* We extend the suboptimality bound from (Ajay et al., 2020) between goal conditioned policies $\pi^*$ and $\pi_{\theta_H}^H$ as follows:

$$|J(\pi^*) - J(\pi_{\theta_H}^H)| \leq \frac{2}{(1-\gamma)(1-\gamma^c)} R_{max} \mathbb{E}_{s \sim d_c^{\pi^*}, g \sim G}[D_{TV}(\pi^*(\tau|s,g)||\pi_{\theta_H}^H(\tau|s,g))] \quad (14)$$

By applying triangle inequality:

$$D_{TV}(\pi^*(\tau|s,g)||\pi_{\theta_H}^H(\tau|s,g)) \leq D_{TV}(\pi^*(\tau|s,g)||\pi_D^H(\tau|s,g)) + D_{TV}(\pi_D^H(\tau|s,g)||\pi_{\theta_H}^H(\tau|s,g)) \quad (15)$$

Taking expectation wrt $s \sim \kappa$, $g \sim G$ and $\pi_D^H \sim \Pi_D^H$,

$$\mathbb{E}_{s \sim \kappa, g \sim G}[D_{TV}(\pi^*(\tau|s,g)||\pi_{\theta_H}^H(\tau|s,g))] \leq \mathbb{E}_{s \sim \kappa, \pi_D^H \sim \Pi_D^H, g \sim G}[D_{TV}(\pi^*(\tau|s,g)||\pi_D^H(\tau|s,g))] +$$
$$\mathbb{E}_{s \sim \kappa, \pi_D^H \sim \Pi_D^H, g \sim G}[D_{TV}(\pi_D^H(\tau|s,g)||\pi_{\theta_H}^H(\tau|s,g))] \quad (16)$$

Since $\pi^*$ is $\phi_D$ common in $\Pi_D^H$, we can write 16 as:

$$\mathbb{E}_{s \sim \kappa, g \sim G}[D_{TV}(\pi^*(\tau|s,g)||\pi_{\theta_H}^H(\tau|s,g))] \leq$$
$$\phi_D + \mathbb{E}_{s \sim \kappa, \pi_D^H \sim \Pi_D^H, g \sim G}[D_{TV}(\pi_D^H(\tau|s,g)||\pi_{\theta_H}^H(\tau|s,g))] \quad (17)$$

Substituting the result from Equation 17 in Equation 14, we get

$$|J(\pi^*) - J(\pi_{\theta_H}^H)| \leq \lambda_H * \phi_D + \lambda_H * \mathbb{E}_{s \sim \kappa, \pi_D^H \sim \Pi_D^H, g \sim G}[D_{TV}(\pi_D^H(\tau|s,g)||\pi_{\theta_H}^H(\tau|s,g))]] \quad (18)$$

where $\lambda_H = \frac{2}{(1-\gamma)(1-\gamma^c)} R_{max} \|\frac{d_c^{\pi^*}}{\kappa}\|_\infty$ □

### A.1.2 SUB-OPTIMALITY PROOF FOR LOWER LEVEL POLICY

Let the optimal lower level policy be $\pi^{**}$. The suboptimality of lower primitive $\pi_{\theta_L}^L$ can be bounded as follows:

$$|J(\pi^{**}) - J(\pi_{\theta_L}^L)| \le \lambda_L * \phi_D + \lambda_L * \mathbb{E}_{s \sim \kappa, \pi_D^L \sim \Pi_D^L, s_g \sim \pi_{\theta_H}^H}[D_{TV}(\pi_D^L(\tau|s,s_g)||\pi_{\theta_L}^L(\tau|s,s_g))]] \tag{19}$$

where $\lambda_L = \frac{2}{(1-\gamma)^2} R_{max} \|\frac{d_c^{\pi^{**}}}{\kappa}\|_\infty$

*Proof.* We extend the suboptimality bound from (Ajay et al., 2020) between goal conditioned policies $\pi^{**}$ and $\pi_{\theta_L}^L$ as follows:

$$|J(\pi^{**}) - J(\pi_{\theta_L}^L)| \le \frac{2}{(1-\gamma)^2} R_{max} \mathbb{E}_{s \sim d_c^{\pi^{**}}, s_g \sim \pi_{\theta_H}^H}[D_{TV}(\pi^{**}(\tau|s,s_g)||\pi_{\theta_L}^L(\tau|s,s_g))] \tag{20}$$

By applying triangle inequality:

$$D_{TV}(\pi^{**}(\tau|s,s_g)||\pi_{\theta_L}^L(\tau|s,s_g)) \le D_{TV}(\pi^{**}(\tau|s,s_g)||\pi_D^L(\tau|s,s_g)) + \\ D_{TV}(\pi_D^L(\tau|s,s_g)||\pi_{\theta_L}^L(\tau|s,s_g)) \tag{21}$$

Taking expectation wrt $s \sim \kappa$, $s_g \sim \pi_{\theta_H}^H$ and $\pi_D^L \sim \Pi_D^L$,

$$\mathbb{E}_{s \sim \kappa, s_g \sim \pi_{\theta_H}^H}[D_{TV}(\pi^{**}(\tau|s,s_g)||\pi_{\theta_L}^L(\tau|s,s_g))] \le \\ \mathbb{E}_{s \sim \kappa, \pi_D^L \sim \Pi_D^L, s_g \sim \pi_{\theta_H}^H}[D_{TV}(\pi^{**}(\tau|s,s_g)||\pi_D^L(\tau|s,s_g))] + \\ \mathbb{E}_{s \sim \kappa, \pi_D^L \sim \Pi_D^L, s_g \sim \pi_{\theta_H}^H}[D_{TV}(\pi_D^L(\tau|s,s_g)||\pi_{\theta_L}^L(\tau|s,s_g))] \tag{22}$$

Since $\pi^{**}$ is $\phi_D$ common in $\Pi_D^L$, we can write 22 as:

$$\mathbb{E}_{s \sim \kappa, s_g \sim \pi_{\theta_H}^H}[D_{TV}(\pi^{**}(\tau|s,s_g)||\pi_{\theta_L}^L(\tau|s,s_g))] \le \\ \phi_D + \mathbb{E}_{s \sim \kappa, \pi_D^L \sim \Pi_D^L, s_g \sim \pi_{\theta_H}^H}[D_{TV}(\pi_D^L(\tau|s,s_g)||\pi_{\theta_L}^L(\tau|s,s_g))] \tag{23}$$

Substituting the result from Equation 23 in Equation 20, we get

$$|J(\pi^{**}) - J(\pi_{\theta_L}^L)| \le \lambda_L * \phi_D + \lambda_L * \mathbb{E}_{s \sim \kappa, \pi_D^L \sim \Pi_D^L, s_g \sim \pi_{\theta_H}^H}[D_{TV}(\pi_D^L(\tau|s,s_g)||\pi_{\theta_L}^L(\tau|s,s_g))]] \tag{24}$$

where $\lambda_L = \frac{2}{(1-\gamma)^2} R_{max} \|\frac{d_c^{\pi^{**}}}{\kappa}\|_\infty$ □

### A.2 GENERATING EXPERT DEMONSTRATIONS

For maze navigation, we use path planning RRT (Lavalle, 1998) algorithm to generate expert demonstration trajectories. For pick and place, we hard coded an optimal trajectory generation policy for generating demonstrations, although they can also be generated using Mujoco VR (Todorov et al., 2012). For kitchen task, the expert demonstrations are collected using Puppet Mujoco VR system (Fu et al., 2020). In rope manipulation task, expert demonstrations are generated by repeatedly finding the closest corresponding rope elements from the current rope configuration and final goal rope configuration, and performing consecutive pokes of a fixed small length on the rope element in the direction of the goal configuration element. The detailed procedure are as follows:

### A.2.1 MAZE NAVIGATION TASK

We use the path planning RRT (Lavalle, 1998) algorithm to generate optimal paths $P = (p_t, p_{t+1}, p_{t+2}, ...p_n)$ from the current state to the goal state. RRT has privileged information about the obstacle position which is provided to the methods through state. Using these expert paths, we generate state-action expert demonstration dataset for the lower level policy.

### A.2.2   Pick and place task

In order to generate expert demonstrations, we can either use a human expert to perform the pick and place task in virtual reality based Mujoco simulation, or hard code a control policy. We hard-coded the expert demonstrations in our setup. In this task, the robot firstly picks up the block using robotic gripper, and then takes it to the target goal position. Using these expert trajectories, we generate expert demonstration dataset for the lower level policy.

### A.2.3   Bin task

In order to generate expert demonstrations, we can either use a human expert to perform the bin task in virtual reality based Mujoco simulation, or hard code a control policy. We hard-coded the expert demonstrations in our setup. In this task, the robot firstly picks up the block using robotic gripper, and then places it in the target bin. Using these expert trajectories, we generate expert demonstration dataset for the lower level policy.

### A.2.4   Hollow task

In order to generate expert demonstrations, we can either use a human expert to perform the hollow task in virtual reality based Mujoco simulation, or hard code a control policy. We hard-coded the expert demonstrations in our setup. In this task, the robotic gripper has to pick up the square hollow block and place it such that a vertical structure on the table goes through the hollow block. Using these expert trajectories, we generate expert demonstration dataset for the lower level policy.

### A.2.5   Rope Manipulation Environment

We hand coded an expert policy to automatically generate expert demonstrations $e = (s_0^e, s_1^e, \ldots, s_{T-1}^e)$, where $s_i^e$ are demonstration states. The states $s_i^e$ here are rope configuration vectors. The expert policy is explained below.

Let the starting and goal rope configurations be $sc$ and $gc$. We find the cylinder position pair $(sc_m, gc_m)$ where $m \in [1, n]$, such that $sc_m$ and $gc_m$ are farthest from each other among all other cylinder pairs. Then, we perform a poke $(x, y, \theta)$ to drag $sc_m$ towards $gc_m$. The $(x, y)$ position of the poke is kept close to $sc_m$, and poke direction $\theta$ is the direction from $sc_m$ towards $gc_m$. After the poke execution, the next pair of farthest cylinder pair is again selected and another poke is executed. This is repeatedly done for $k$ pokes, until either the rope configuration $sc$ comes within $\delta$ distance of goal $gc$, or we reach maximum episode horizon $T$. Although, this policy is not the perfect policy for goal based rope manipulation, but it still is a good expert policy for collecting demonstrations $\mathcal{D}$. Moreover, as our method requires states and not primitive actions (pokes), we can use these demonstrations $\mathcal{D}$ to collect good higher level subgoal dataset $\mathcal{D}_g$ using primitive parsing.

### A.3   Environment implementation details

Here, we provide extensive environment and implementation details for various environments. The experiments are run for $4.73e5$, $1.1e5$, $1.32E5$, $1.8E5$, $1.58e6$, and $5.32e5$ timesteps in maze, pick and place, bin, hollow, rope and kitchen respectively. The regularization weight hyper-parameter $\Psi$ is set at 0.001, 0.005, 0.005, 0.005, 0.005, and 0.005, the population hyper-parameter $p$ is set to be $1.1e4$, 2500, 2500, 2500, $3.9e5$, and $1.4e4$, and distance threshold hyper-parameter $Q_{thresh}$ is set at 10, 0, 0, 0, 0, and 0 for maze, pick and place, bin, hollow, rope and kitchen tasks respectively. In maze navigation, a 7-DOF robotic arm navigates across randomly generated four room mazes, where the closed gripper (fixed at table height) has to navigate across the maze to the goal position. In pick and place task, the 7-DOF robotic arm gripper has to navigate to the square block, pick it up and bring it to the goal position. In bin task, the 7-DOF robotic arm gripper has to pick the square block and place the block inside the bin. In hollow task, the 7-DOF robotic arm gripper has to pick a square hollow block and place it such that a fixed vertical structure on the table goes through the hollow block. In rope manipulation task, a deformable soft rope is kept on the table and the 7-DoF robotic arm performs pokes to nudge the rope towards the desired goal rope configuration. The rope manipulation task involves learning challenging dynamics and goes beyond prior work on navigation-like tasks where the goal space is limited. In the kitchen task, the 9-DoF franka robot has to perform a complex multi-stage task in order to achieve the final goal. Although many

such permutations can be chosen, we formulate the following task: the robot has to first open the microwave door, then switch on the specific gas knob where the kettle is placed.

In maze navigation, upper level predicts a subgoal, and the lower level primitive travels in a straight line towards the predicted goal. In pick and place, bin and hollow tasks, we design three primitives, gripper-reach: where the gripper goes to given position $(x_i, y_i, z_i)$, gripper-open: opens the gripper, and gripper-close: closes the gripper. In kitchen environment, we use the action primitives implemented in RAPS (Dalal et al., 2021). While using RAPS baseline, we hand designed action primitives, which we provide in detail in Section A.3.

### A.3.1 MAZE NAVIGATION TASK

In this environment, a 7-DOF robotic arm gripper navigates across random four room mazes. The gripper arm is kept closed and the positions of walls and gates are randomly generated. The table is discretized into a rectangular $W * H$ grid, and the vertical and horizontal wall positions $W_P$ and $H_P$ are randomly picked from $(1, W - 2)$ and $(1, H - 2)$ respectively. In the four room environment thus constructed, the four gate positions are randomly picked from $(1, W_P - 1)$, $(W_P + 1, W - 2)$, $(1, H_P - 1)$ and $(H_P + 1, H - 2)$. The height of gripper is kept fixed at table height, and it has to navigate across the maze to the goal position(shown as red sphere).

The following implementation details refer to both the higher and lower level polices, unless otherwise explicitly stated. The state and action spaces in the environment are continuous. The state is represented as the vector $[p, \mathcal{M}]$, where $p$ is current gripper position and $\mathcal{M}$ is the sparse maze array. The higher level policy input is thus a concatenated vector $[p, \mathcal{M}, g]$, where $g$ is the target goal position, whereas the lower level policy input is concatenated vector $[p, \mathcal{M}, s_g]$, where $s_g$ is the sub-goal provided by the higher level policy. The current position of the gripper is the current achieved goal. The sparse maze array $\mathcal{M}$ is a discrete $2D$ one-hot vector array, where 1 represents presence of a wall block, and 0 absence. In our experiments, the size of $p$ and $\mathcal{M}$ are kept to be 3 and 110 respectively. The upper level predicts subgoal $s_g$, hence the higher level policy action space dimension is the same as the dimension of goal space of lower primitive. The lower primitive action $a$ which is directly executed on the environment, is a 4 dimensional vector with every dimension $a_i \in [0, 1]$. The first 3 dimensions provide offsets to be scaled and added to gripper position for moving it to the intended position. The last dimension provides gripper control(0 implies a fully closed gripper, 0.5 implies a half closed gripper and 1 implies a fully open gripper). We select 100 randomly generated mazes each for training, testing and validation. For selecting train, test and validation mazes, we first randomly generate 300 distinct mazes, and then randomly divide them into 100 train, test and validation mazes each. We use off-policy Soft Actor Critic (Haarnoja et al., 2018b) algorithm for optimizing RL objective in our experiments.

### A.3.2 PICK AND PLACE, BIN AND HOLLOW ENVIRONMENTS

In the pick and place environment, a 7-DOF robotic arm gripper has to pick a square block and bring/place it to a goal position. We set the goal position slightly higher than table height. In this complex task, the gripper has to navigate to the block, close the gripper to hold the block, and then bring the block to the desired goal position. In the bin environment, the 7-DOF robotic arm gripper has to pick a square block and place it inside a fixed bin. In the hollow environment, the 7-DOF robotic arm gripper has to pick a hollow plate from the table and place it on the table such that its hollow center goes through a fixed vertical pole placed on the table. In all the three environments, the state is represented as the vector $[p, o, q, e]$, where $p$ is current gripper position, $o$ is the position of the block object placed on the table, $q$ is the relative position of the block with respect to the gripper, and $e$ consists of linear and angular velocities of the gripper and the block object. The higher level policy input is thus a concatenated vector $[p, o, q, e, g]$, where $g$ is the target goal position. The lower level policy input is concatenated vector $[p, o, q, e, s_g]$, where $s_g$ is the sub-goal provided by the higher level policy. The current position of the block object is the current achieved goal. In our experiments, the sizes of $p$, $o$, $q$, $e$ are kept to be 3, 3, 3 and 11 respectively. The upper level predicts subgoal $s_g$, hence the higher level policy action space and goal space have the same dimension. The lower primitive action $a$ is a 4 dimensional vector with every dimension $a_i \in [0, 1]$. The first 3 dimensions provide gripper position offsets, and the last dimension provides gripper control (0 means closed gripper and 1 means open gripper). While training, the position of block object and goal are randomly generated (block is always initialized on the table, and goal is always above the

table at a fixed height). We select 100 random each for training, testing and validation. For selecting train, test and validation mazes, we first randomly generate 300 distinct environments with different block and target goal positions, and then randomly divide them into 100 train, test and validation mazes each. We use off-policy Soft Actor Critic (Haarnoja et al., 2018b) algorithm for the RL objective in our experiments.

### A.3.3 ROPE MANIPULATION ENVIRONMENT

In the robotic rope manipulation task, a deformable rope is kept on the table and the robotic arm performs pokes to nudge the rope towards the desired goal rope configuration. The task horizon is fixed at 25 pokes. The deformable rope is formed from 15 constituent cylinders joined together. The following implementation details refer to both the higher and lower level polices, unless otherwise explicitly stated. The state and action spaces in the environment are continuous. The state space for the rope manipulation environment is a vector formed by concatenation of the intermediate joint positions. The upper level predicts subgoal $s_g$ for the lower primitive. The action space of the poke is $(x, y, \eta)$, where $(x, y)$ is the initial position of the poke, and $\eta$ is the angle describing the direction of the poke. We fix the poke length to be $0.08$. While training our hierarchical approach, we select 100 randomly generated initial and final rope configurations each for training, testing and validation. For selecting train, test and validation configurations, we first randomly generate 300 distinct configurations, and then randomly divide them into 100 train, test and validation mazes each. We use off-policy Soft Actor Critic (Haarnoja et al., 2018b) algorithm for optimizing RL objective in our experiments.

### A.4 ABLATION EXPERIMENTS

Here, we present the ablation experiments in all four task environments. The ablation analysis includes experiments to choose RPL window size $k$ hyperparameter (Figure 5), learning weight hyperparameter $\phi$ (Figure 6), comparison between PEAR-IRL, PEAR-BC and PEAR-RPL ablation (Figure 7), and comparisons with varying number of expert demonstrations used during relabeling and training (Figure 8).

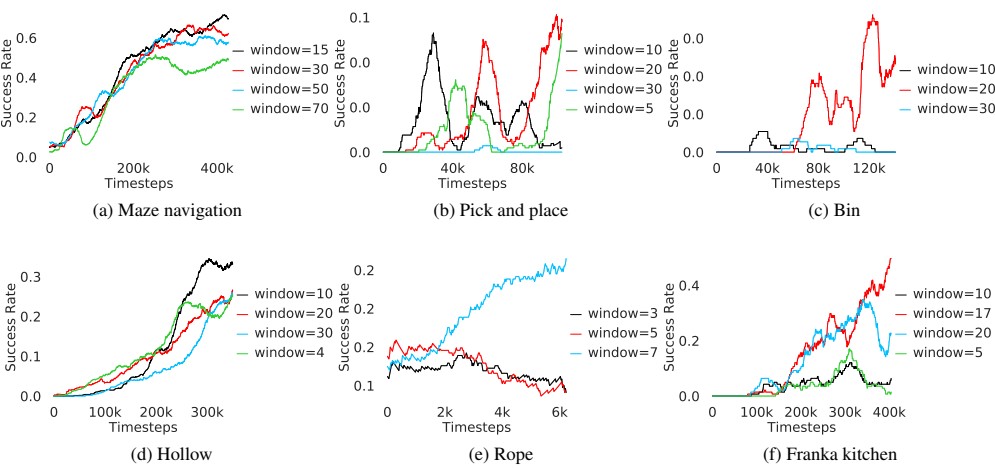

Figure 5: The success rate plots show the performance of RPL for values of $k$ window size parameter versus number of training epochs.

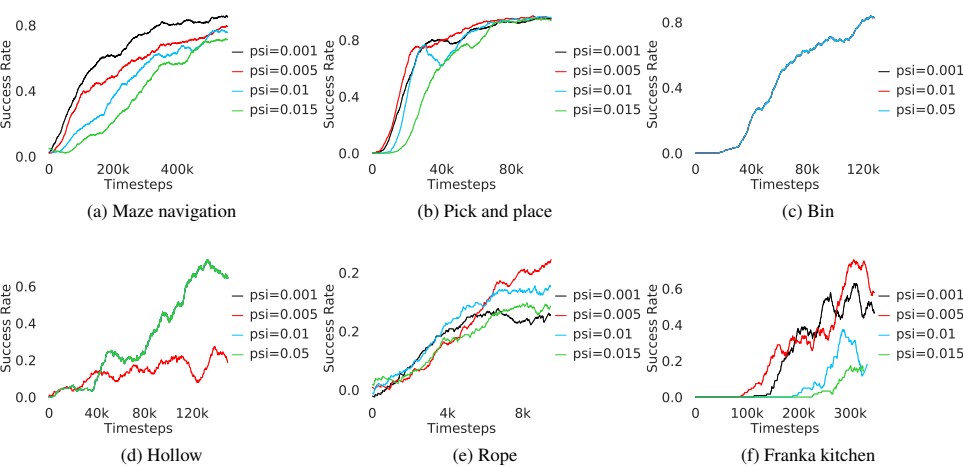

Figure 6: The success rate plots show performance of CRISP for values of learning weight parameter $\psi$ versus number of training timesteps.

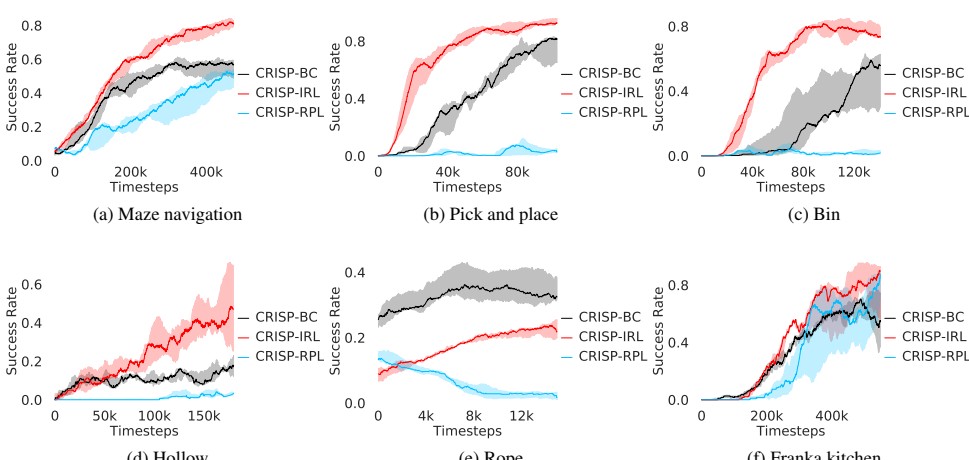

Figure 7: The success rate plots show success rate performance comparison between PEAR-IRL, PEAR-BC and PEAR-RPL ablation.

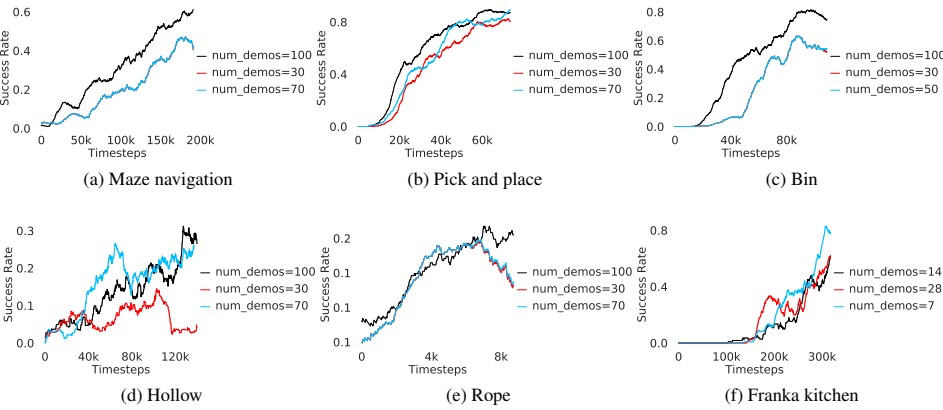

Figure 8: The success rate plots show success rate performance plots of varying number of expert demonstrations versus number of training epochs.

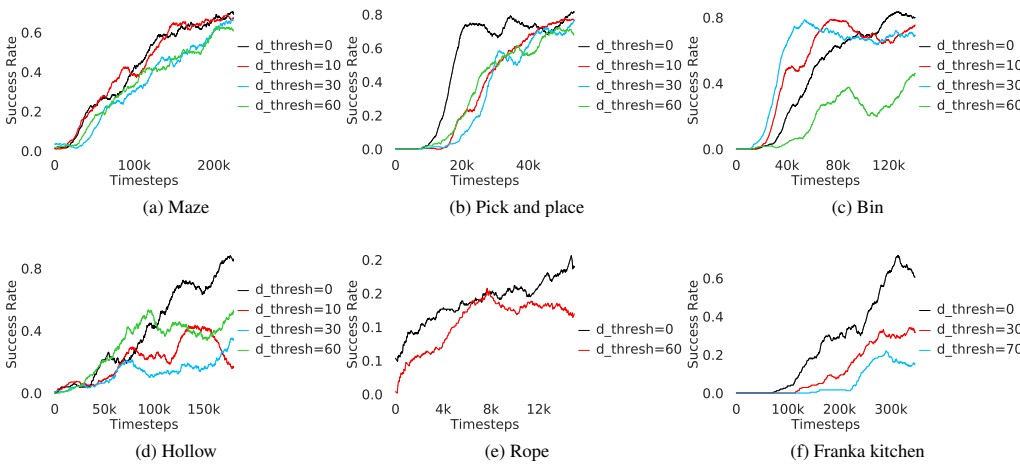

Figure 9: The success rate plots show the performance of CRISP for various values of $Q_{thresh}$ parameter versus number of training timesteps.

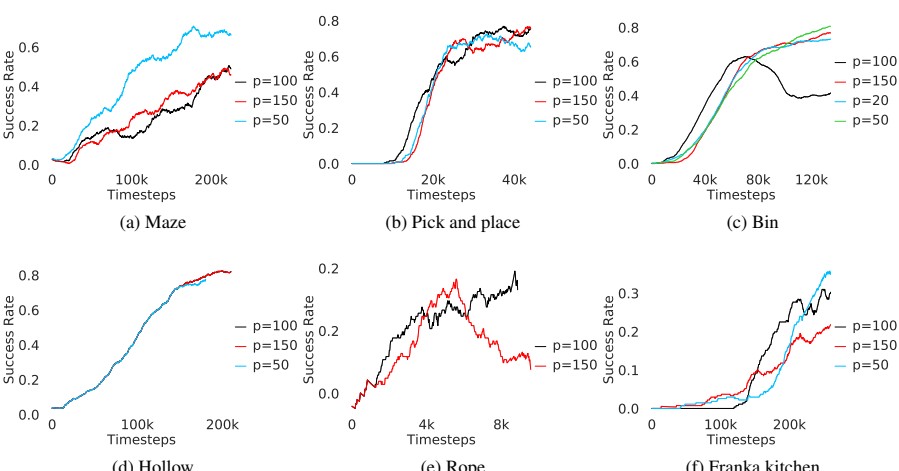

Figure 10: The success rate plots show the performance of CRISP for various values of population number $p$ parameter versus number of training timesteps.

## A.5 QUALITATIVE VISUALIZATIONS

In this subsection, we provide visualization of successful and failure cases for some of the testing runs for various environments in Figures 10-19:

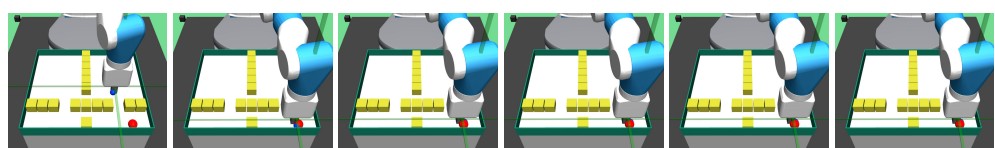

Figure 12: **Successful visualization**: The visualization is a successful attempt at performing maze navigation task

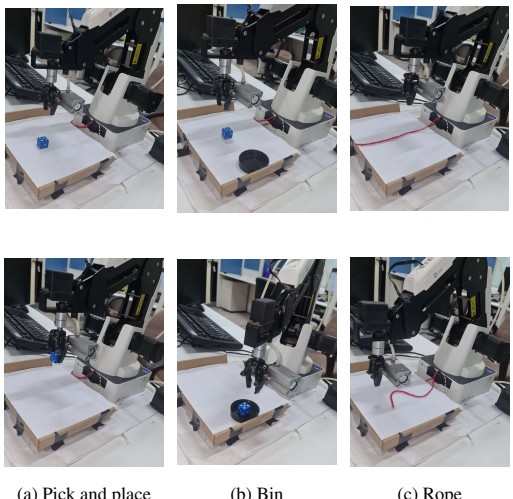

| (a) Pick and place | (b) Bin | (c) Rope |

Figure 11: **Real world experiments** in pick and place, bin and rope manipulation environments. Row 1 depicts initial and Row 2 depicts goal configuration.

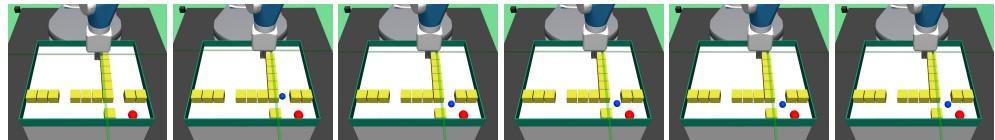

Figure 13: **Failed visualization**: The visualization is a failed attempt at performing maze navigation task

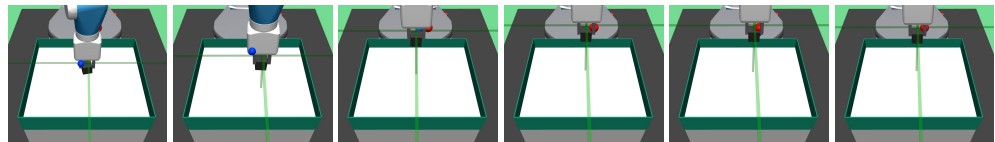

Figure 14: **Successful visualization**: The visualization is a successful attempt at performing pick navigation task

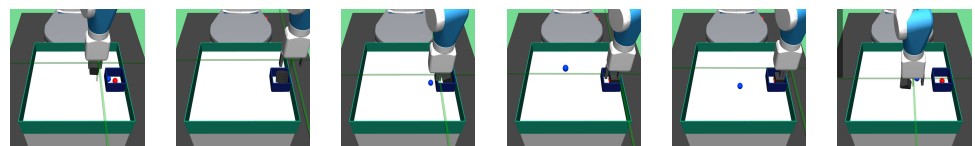

Figure 15: **Successful visualization**: The visualization is a successful attempt at performing bin task

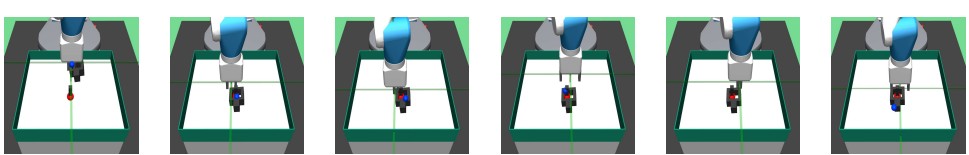

Figure 16: **Successful visualization**: The visualization is a successful attempt at performing hollow task

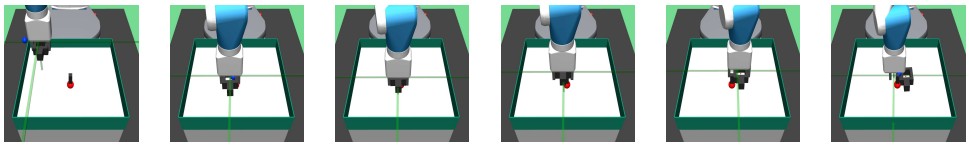

Figure 17: **Failed visualization**: The visualization is a falied attempt at performing hollow task

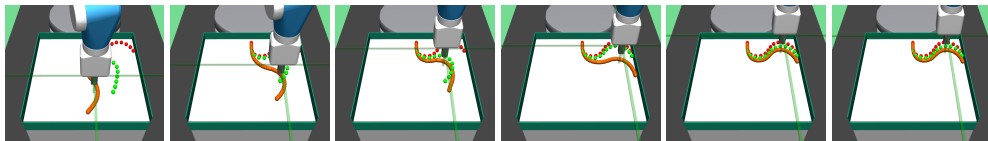

Figure 18: **Successful visualization**: The visualization is a successful attempt at performing rope navigation task

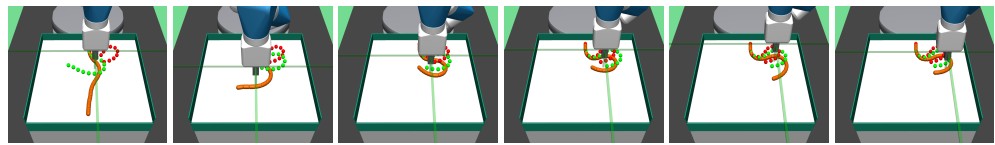

Figure 19: **Failed visualization**: The visualization is a failed attempt at performing rope navigation task

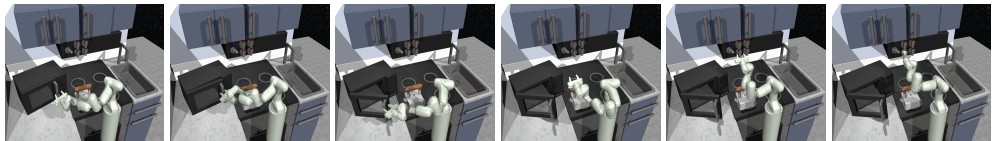

Figure 20: **Successful visualization**: The visualization is a successful attempt at performing kitchen navigation task

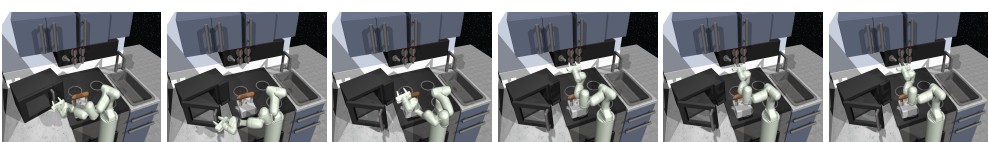

Figure 21: **Failed visualization**: The visualization is a failed attempt at performing kitchen navigation task

