# OpenReview forum: "PEAR: Primitive enabled Adaptive Relabeling for boosting Hierarchical Reinforcement Learning"
_ICLR.cc/2024/Conference — Submitted to ICLR 2024_

### Official Review · Reviewer_hXvK · 2023-10-30

**Soundness:** 3 good
**Presentation:** 3 good
**Contribution:** 3 good
**Rating:** 6
**Confidence:** 3

**Summary:**

The authors present PEAR, which first relabels expert demonstrations to obtain a more effective set of subgoals, and then optimizes a hierarchical reinforcement learning (HRL) agent using reinforcement learning and imitation learning on the relabeled expert demonstrations. The authors theoretically and experimentally validate their approach.

**Strengths:**

- The authors present a nice and simple idea that performs well.
- The authors theoretically validate their relabeling algorithm in the appendix.
- The authors demonstrate their algorithm's performance on an extensive set of tasks, including on real world robot tasks.
- The authors qualitatively show the subgoal predictions in Figure 2.

**Weaknesses:**

- The learning curve figures are too small and very difficult to see. It would be much appreciated if the authors could fix this.
- In Algorithm 1 for Lines 5-13 it could improve clarity if the authors included comments on what each line is doing. I found this part a bit difficult to understand.
- It seems it is necessary to manually set $Q_{\text{thresh}}$ for each environment.

**Questions:**

- I'm curious why PEARL-IRL outperforms PEAR_BC on Maze, Pick Place, Bin, and Hollow, but PEAR-BC outperforms PEARL-IRL on Kitchen. Do the authors have any ideas why this is?
- Have the authors experimented with different numbers of expert demonstrations? I'm curious how the method would perform with more/less demonstrations.
- How is $Q_{\text{thresh}}$ chosen for each environment?

---

> ### Author Response · Authors · 2023-11-17
> **Author Response:**
>
> Reviewer 4:
>
> We thank the reviewer for their detailed, constructive feedback. We address the reviewer’s concerns as follows:
>
> **1. "The learning curve figures are too small and very difficult to see. It would be much appreciated if the authors could fix this."**
>
> Response:
>
> We apologize for the figure size and clarity issues. We have now fixed the issues in the paper. We hope this improves the overall clarity.
>
> **2. "In Algorithm 1 for Lines 5-13 it could improve clarity if the authors included comments on what each line is doing. I found this part a bit difficult to understand."**
>
> Response:
>
> We thank the reviewer for pointing this out. We have now added the comments in Algorithm 1 wherever applicable. We hope that this improves the clarity of the algorithm psuedo-code.
>
> **3. "It seems it is necessary to manually set $Q_{thresh}$ for each environment."**
>
> Response:
>
> The proposed approach does require setting $Q_{thresh}$ hyper-parameter but we empirically found that the performance is not unstable with varying hyperparameter value. Intuitively, this is a reasonable trade-off that renders the higher level subgoal generation a self-supervised process, thereby eliminating the need for expert guided annotations for segmenting subgoals. Additionally, please note that we empirically found that after we normalize $Q_{\pi^L}$ values of a trajectory, the $Q_{thresh}$ hyper-parameter value of $0$ works consistently well across all environments, which exhibits that setting $Q_{thresh}$ value does not cause significant overhead. We have added the hyper-parameter ablation analysis in Appendix A.4 Figure 10 for your consideration. Additionally, in real robotic experiments, we used the default $Q_{thresh}$ value of $0$ and PEAR was able to outperform the baselines in all experiments.
>
> **4. "I'm curious why PEARL-IRL outperforms PEAR_BC on Maze, Pick Place, Bin, and Hollow, but PEAR-BC outperforms PEARL-IRL on Kitchen. Do the authors have any ideas why this is?"**
>
> Response:
>
> Our motivation for using inverse reinforcement learning (IRL) is that although behavior cloning (BC) works efficiently in many scenarios, it sometimes fails to perform well for complex tasks that require long term planning. However, the advantages of IRL are sometimes counterbalanced by the fact that they are really difficult to train. We believe that since it is a really difficult environment to train on, PEAR-BC slightly outperforms PEAR-IRL in Franka kitchen environment:
>
> Success Rates:
> | Method    | Success Rate |
> | -------- | ------- |
> | PEAR-IRL  | **0.89 $\pm$ 0.06**   |
> | PEAR-BC | **1.0 $\pm$ 0.0**    |
>
> Thus, although in general, PEAR-IRL works better than PEAR-BC in long horizon complex tasks, PEAR-BC might work better in some cases where it is hard to train an IRL objective. However, the performance does not differ significantly and we found that in real world tasks, PEAR-IRL consistently outperforms PEAR-BC in the experiments.
>
> **5. "Have the authors experimented with different numbers of expert demonstrations? I'm curious how the method would perform with more/less demonstrations."**
>
> Response:
>
> Yes, we have performed ablations to analyze the effect of varying the number of expert demonstrations for each task (Please refer to Appendix A.4 Figure 9). Although it is subject to availability, we increase the number of expert data until there is no significant performance boost.
>
> **6. "How is $Q_{thresh}$ chosen for each environment?"**
>
> Response:
>
> Firstly, we select $100$ randomly generated environments for training, testing and validation. Using this, for calculating $Q_{thresh}$, we compute success rate plots for various $Q_{thresh}$ values to select the best $Q_{thresh}$ parameter (the ablation analysis is shown in Appendix A.4 Figure 10).
>
> We hope that the response addresses the reviewer’s concern. Please let us know, and we will be happy to address additional concerns if any.

---

> > ### Author Response · Authors · 2023-11-21
> > **Discussion:**
> >
> > Dear reviewer,
> >
> > We hope our response clarified your initial concerns/questions. We would be happy to provide further clarifications where necessary.

---

> > > ### Comment · Reviewer_hXvK · 2023-11-23
> > >
> > > Thanks for the changes and the discussion! For Algorithm 1, I see that you have added comments; it would improve clarity if you prepended a # or // to each comment line. Otherwise, it currently blends in with the actual pseudocode. I also appreciate that you have made the learning curve figures in the main text more clear, but the learning curves in the Appendix could also be made much more clear (although these changes will not affect my score).

---

> > > > ### Author Response · Authors · 2023-11-23
> > > > **Author Response:**
> > > >
> > > > We would like to thank the reviewer for taking the time to review the paper and provide very helpful comments. We have added the // to comments and improved the learning curves in the Appendix as advised. Thank you

---

### Official Review · Reviewer_cZN6 · 2023-10-30

**Soundness:** 2 fair
**Presentation:** 1 poor
**Contribution:** 1 poor
**Rating:** 3
**Confidence:** 4

**Summary:**

This paper introduces an approach that conducts adaptive re-labeling on a handful of expert demonstrations to address complex long-horizon tasks in hierarchical reinforcement learning. It offers a bound for the method's suboptimality. The proposed method is experimented on simulated tasks and surpasses several hierarchical reinforcement learning benchmarks.

**Strengths:**

The paper introduces a simple method for generating the subgoal dataset from demonstrations and merges existing solutions for HRL training. By using ower-level policy to adaptively segment expert state-demonstrations into skills, the method provides an appealing feature of replacing the expert annotation. The paper also provides the bounds for the suboptimality of both the higher-level and lower-level policies.

**Weaknesses:**

The core idea of generating the subgoal by thresholding an environment-specific value has natural constraints. The method of determining reachability via the low-level Q function concurs the approach in [*]. In optimizing the hierarchical policies, the paper echoes [**] by generating achievable subgoals through adversarial learning on relabeled subgoals in the HRL context. The authors are encouraged to provide a more in-depth discussion about the novelty of their method and its distinction from existing literature. The result figures provided are too small to interpret.

[*] Kreidieh, Abdul Rahman, et al. "Inter-level cooperation in hierarchical reinforcement learning." arXiv preprint arXiv:1912.02368 (2019).
[*] Wang, et al. "State-conditioned adversarial subgoal generation." Proceedings of the AAAI Conference on Artificial Intelligence. Vol. 37. No. 8. 2023.

**Questions:**

See the above section

---

> ### Author Response · Authors · 2023-11-17
> **Author Response 1/2:**
>
> Reviewer 3:
>
> We thank the reviewer for their detailed, constructive feedback. We address the reviewer’s concerns as follows:
>
> **1. "The core idea of generating the subgoal by thresholding an environment-specific value has natural constraints."**
>
> Response:
>
> The proposed approach does require setting $Q_{thresh}$ hyper-parameter but we empirically found that the performance is not unstable with varying hyperparameter value. Intuitively, this is a reasonable trade-off that renders the higher level subgoal generation a self-supervised process, thereby eliminating the need for expert guided annotations for segmenting subgoals. Additionally, please note that we empirically found that after we normalize $Q_{\pi^L}$ values of a trajectory, the $Q_{thresh}$ hyper-parameter value of $0$ works consistently well across all environments, which exhibits that setting $Q_{thresh}$ value does not cause significant overhead. We have added the hyper-parameter ablation analysis in Appendix A.4 Figure 10 for your consideration. Additionally, in real robotic experiments, we used the default $Q_{thresh}$ value of $0$ and PEAR was able to outperform the baselines in all experiments.
>
> **2. "The method of determining reachability via the low-level Q function concurs the approach in [Inter-level cooperation in hierarchical reinforcement learning.]"**
>
> Response:
>
> The paper titled "Inter-level cooperation in hierarchical reinforcement learning" proposes an approach based on inter-level cooperation between multi-level agents by jointly optimizing the higher level and lower level functions using RL. Similar to our approach, the paper considers the lower level Q function to motivate the higher level agent to produce reachable subgoals for the lower level policy. However in their approach, the subgoals are encountered as a result of environment exploration of the higher level policy, which may be quite random. This leads to two major issues: (i) either the generated subgoals are very hard, which impedes effective lower level policy learning, or (ii) the subgoals are too easy, due to which the higher level has to do all the heavy lifting to solve the task. The issue is exacerbated in the beginning of training, due to which the policies have to extensively explore the environment before being able to come across good subgoal predictions. Due to these issues, the approach might be unable to perform well on multi-stage long horizon tasks.
> In our approach, we use a handful of expert demonstrations and employ our novel "primitive enabled adaptive relabeling" procedure for picking reachable subgoals for the lower level policy, and effectively utilize them for training higher level policy using additional imitation learning regularization. Notably, expert demonstrations can not be directly leveraged in raw trajectory form to train the higher level, and therefore require efficient data relabeling to segment them into meaningful subgoals. Moreover, since we periodically perform data relabeling, our approach always generates reachable subgoals for lower level policy, thereby mitigating non-stationarity issue prevalent in vanilla hierarchical learning approaches, and thus devising a practical HRL algorithm. In order to prevent over-estimation of the values on out-of-distribution states, we also employ an additional margin classification objective as explained in detail in Section 4.1. We also provide theoretical guarantees that justify the benefits of periodic re-population using adaptive relabeling, which are largely missing in recent contemporary work. We also demonstrate the efficacy of our approach in complex long horizon tasks like Franka kitchen environment, and also include environments with challenging dynamics and policy, especially the soft rope manipulation, which goes beyond some peer work that focuses on navigation-like tasks on 2D plane, where the goal space is limited. Finally, we demonstrate that PEAR is able to demonstrate good performance in challenging real world environments. We have improved the related work section based on the reviwer's concern and the discussion above, and added a brief discussion summarizing above points in the the paper.

---

> > ### Author Response · Authors · 2023-11-17
> > **Author Response 2/2:**
> >
> > **3. " In optimizing the hierarchical policies, the paper echoes [State-conditioned adversarial subgoal generation] by generating achievable subgoals through adversarial learning on relabeled subgoals in the HRL context."**
> >
> > Response:
> >
> > The paper titled "State-conditioned adversarial subgoal generation" proposes an approach which adversarially guides the higher level policy to produce subgoals according to the current instantiation of the lower level policy. Although this approach deals with non-stationarity, it unfortunately also suffers from possible degenerate solutions, as explained above. Moreover, they require additional assumptions on the subgoal space when relabeling transitions using the approach proposed in HIRO, which may not be satisfied in challenging robotic environments that require solving complex multi-stage tasks (e.g in our novel rope manipulation environment). As explained above, we side-step such issues by adaptively relabeling a handful of expert demonstrations. Additionally, since our approach performs adaptive relabeling periodically, we are able to effectively deal with the non-stationarity issue and demonstrate strong results on complex multi-stage tasks as shown in Section 5. We also provide theoretical bounds and propose a generalized plug-and-play framework for joint optimization using reinforcement learning and imitation learning, where using Kullback–Leibler divergence leads to behavior cloning and Jensen-Shannon divergence leads to inverse reinforcement learning objectives.
> > Additionally, we would like to point out that this work is actually concurrent to our work, as our work was in submission when this work was published. Based on previous reviews, we have added experiments in harder complex environments, added multiple hierarchical and non hierarchical baselines, and added additional margin classification objective for preventing over-estimation on out-of distribution states.
> >
> > **4. " The authors are encouraged to provide a more in-depth discussion about the novelty of their method and its distinction from existing literature."**
> >
> > Response:
> >
> > We thank the reviewer for the stern criticism and helpful suggestions. As explained in detail above above, we have tried to show that this work is indeed novel and distinct from previously proposed approaches, and significantly improves upon existing approaches, and we therefore imply that it be considered for conference submission. Our research ideology is that even though sometimes the inherent ideas for an approach may be very simple and even very similar to previously proposed approaches, they are worth pursuing if they solve the inherent issues and/or add to explainability of the considered research problem, thus paving the way for improved research in that area in the future.
> >
> > **5. "The result figures provided are too small to interpret."**
> >
> > Response:
> >
> > We apologize for the figure size and clarity issues. We have now fixed the issues in the paper. We hope this improves the overall clarity.
> >
> > We hope that the response addresses the reviewer’s concern. Please let us know, and we will be happy to address additional concerns if any.

---

> > > ### Author Response · Authors · 2023-11-21
> > > **Discussion:**
> > >
> > > Dear reviewer,
> > >
> > > We hope our response clarified your initial concerns/questions. We would be happy to provide further clarifications where necessary.

---

> ### Comment · Reviewer_cZN6 · 2023-11-22
>
> I recognize and appreciate the authors' efforts to address the initial concerns raised. However, my reservations regarding the distinction between this paper and the prior work "State-conditioned adversarial subgoal generation" as well as the environment-specific hyper-parameter remain.
>
> The response provided does not fully clarify how this work significantly diverges from the aforementioned prior work, particularly in terms of optimizing hierarchical policies. While the method of generating relabeled subgoals might differ, the core concept of using adversarial learning in the HRL context to generate achievable subgoals seems to align closely with the previous work. The novelty of this paper appears to hinge on the specific method of generating relabeled subgoals, but this alone may not constitute a substantial advancement.
>
> Furthermore, my concern regarding the method of generating subgoals by thresholding an environment-specific value has not been adequately addressed. Without more comprehensive evaluations, it's challenging to assess the potential limitations or strengths of this approach. This aspect is crucial for understanding the practical applicability and robustness of the proposed method in diverse scenarios.

---

> ### Author Response · Authors · 2023-11-22
> **Author Response:**
>
> 1. **The response provided does not fully clarify how this work significantly diverges from the aforementioned prior work, particularly in terms of optimizing hierarchical policies**
>
> Response:
>
> We understand the concern and provide clarifications to differentiate our method from the aforementioned work. Although our approach and the mentioned work both use adversarial training to generate achievable subgoals, our approach differs from the previous work in the following manner:
>
> 1. As explained before, the approach in the mentioned paper may yield degenerate solutions. However, we side-step this issue by using a handful of expert demonstrations for picking reachable subgoals for the lower level policy, and effectively utilizing them for training higher level policy using additional imitation learning regularization.
>
> 2. The relabeling approach in the mentioned paper relies on the approach in the paper HIRO, which assumes additional assumptions on the environment subgoal generation. In contrast, our approach assumes no such limiting assumptions.
>
> 3. We propose a hierarchical curriculum learning based approach, which always generates a curriculum of achievable subgoals for the lower primitive.
>
> 4. Apart from adversarial training, we derive a general imitation learning based regularization objective, where we can plug in various distance metrics to yield various imitation learning algorithms. We perform theoretical and empirical analysis for inverse reinforcement learning and behavior cloning regularization objectives.
>
> 5. Our approach provides a plug-and-play framework where we can plug in off-policy RL algorithm and IL algorithm of choice, which ultimately yields various joint optimization objectives for hierarchical reinforcement learning.
>
> 6. We also provide theoretical guarantees that justify the benefits of periodic re-population using adaptive relabeling, which is missing from the mentioned work.
>
> 7. We also demonstrate the efficacy of our approach in complex long horizon tasks, and also include environments with challenging dynamics and policy, which goes beyond some peer work where the goal space is constrained.
>
> 8. Additionally, we would like to point out that this work is actually concurrent to our work, as our work was in submission when this work was published.
>
> We hope that the above mentioned points differentiate our work from the previous work, and elucidates the contributions of our approach.
>
>
>
> 2. **Furthermore, my concern regarding the method of generating subgoals by thresholding an environment-specific value has not been adequately addressed**
>
> Response:
>
> In our approach, $Q_{thresh}$ is used for automatically relabeling expert demonstrations to generate expert subgoal dataset for training higher level policies. Some prior works use fixed window based relabeling to generate expert subgoal demonstration dataset for the higher level policy [https://arxiv.org/abs/1910.11956]. In such approaches, the window size parameter $k$ is an environment specific value which has to be set. Moreover, this approach is basically a brute force approach where we consider multiple window sizes, and select the best window size from among them. However, intuitively, such approaches suffer from two reasons: (i) it requires environment specific window size hyper-parameter to be set, and (ii) fixed parsing approaches may generate sub-optimal expert subgoal dataset. Although our method uses environment specific $Q_{thresh}$ value, it side-steps the fixed parsing issue by using primitive enabled adaptive relabeling, which leads to better policies learnt from adaptively relabeled expert demonstrations. Moreover, our higher level subgoal generation process (primitive enabled adaptive relabeling) is a self-supervised process, which does not require an expert for segmenting subgoals.
>
> Additionally, please note that the $Q_{thresh}$ hyper-parameter value of $0$ works consistently well across all environments, which demonstrates that setting $Q_{thresh}$ value does not cause significant overhead. Thus, we effectively just need to normalize $Q_{\pi^L}$ values of a trajectory, and then select $Q_{thresh}$ hyper-parameter value of $0$, which works consistently well across all environments.
>
> We hope that the response addresses the reviewer’s concerns and clearly explains its contributions.

---

### Official Review · Reviewer_uEcd · 2023-10-31

**Soundness:** 2 fair
**Presentation:** 2 fair
**Contribution:** 2 fair
**Rating:** 6
**Confidence:** 3

**Summary:**

This paper introduces a novel joint optimization approach to address non-stationarity in hierarchical reinforcement learning. By comparing the Q-values of the low-level policy with the environment-specific Q-value, adaptive relabeling of expert samples is performed for both high- and low-level policy training. The authors claim that this approach can mitigate the non-stationarity in HRL. Additionally, the authors provide performance difference bounds under the $\phi_D$ common policy assumption, and simulation as well as real-world experiments demonstrate the comparable performance of PEAR.

**Strengths:**

I think it is a new approach, and this paper provides some theoretical explanations. It also includes experiments in real-world environments.

**Weaknesses:**

1. **[Deal with the non-stationarity]** I don't agree that this work ameliorates non-stationarity. Since PEAR can access expert samples when training the high-level policy by BC or IRL, the high-level dynamics remain consistent with the dynamics that generate the expert sample trajectories without any change. This is inconsistent with the claim of non-stationarity made by the authors, which raises concerns about the contribution in this regard.
2. **[Performance limitation from expert]** Since the high-level policy is optimized only through BC or IRL using expert samples and does not participate in the environment exploration process, the performance of expert samples significantly limits the upper-performance limit of PEAR. Therefore, the authors need to analyze the impact of expert performance on the upper-performance limit of PEAR and conduct experiments with expert samples of varying performance.

3. **[How to get $Q_{thresh}$]** When performing adaptive relabeling, the authors require additional environment-specific Q values, which are not available in standard experimental settings and may limit the practicality of PEAR.
4. Many of the algorithm designs are not well supported:
   * Why does the additional margin classification objective in low-level policy optimization prevent over-estimation?
   * In equation (2), why does the low-level policy need the BC regularization objective? In my view, the goal of the low-level policy is to reach sub-goals more effectively, and this regularization is confusing.
   * Why is the joint value function formulated as a summation of $J$ and $J_{BC}$?
5. The clarity of the figures in the experiments is poor, especially in Figure 5.

**Questions:**

1. In section 3, the authors define the high-level reward, $r_{ex}$. However, the subsequent high-level policy optimization process does not appear to utilize $r_{ex." Please clarify the purpose of the high-level reward.

2. I recommend that the authors survey and include some of the most recent works in the related works section.

   [1] Lee S, Kim J, Jang I, et al. DHRL: A Graph-Based Approach for Long-Horizon and Sparse Hierarchical Reinforcement Learning[J]. Advances in Neural Information Processing Systems, 2022, 35: 13668-13678.

   [2] Chane-Sane E, Schmid C, Laptev I. Goal-conditioned reinforcement learning with imagined subgoals[C]//International Conference on Machine Learning. PMLR, 2021: 1430-1440.

   [3] Zhang T, Guo S, Tan T, et al. Generating adjacency-constrained subgoals in hierarchical reinforcement learning[J]. Advances in Neural Information Processing Systems, 2020, 33: 21579-21590.

   [4] C-Learning: Learning to Achieve Goals via Recursive Classification

3. In the experimental setup, since PEAR requires training with expert samples (possibly high-performance expert samples), while other baselines like HAC do not depend on expert data, please clarify the fairness of the experimental comparisons.

---

> ### Author Response · Authors · 2023-11-17
> **Author Response 1/2:**
>
> Reviewer 2:
>
> We thank the reviewer for their detailed, constructive feedback. We address the reviewer's concerns as follows:
>
> **1. "[Deal with the non-stationarity] I don't agree that this work ameliorates non-stationarity. Since PEAR can access expert samples when training the high-level policy by BC or IRL, the high-level dynamics remain consistent with the dynamics that generate the expert sample trajectories without any change. This is inconsistent with the claim of non-stationarity made by the authors, which raises concerns about the contribution in this regard."**
>
> Response:
>
> We understand the reviewer's confusion and explain the claim regarding non-stationarity as follows: as explained in detail in Section 4, PEAR jointly optimizes HRL agents by employing reinforcement learning (RL) with imitation learning (IL) regularization. Since the higher level policy does not have direct access to expert subgoal dataset, PEAR using primitive enabled adaptive relabeling to generate efficient subgoal supervision for the higher level policy. While PEAR leverages imitation learning regularization using expert demonstrations to bootstrap learning, it also employs reinforcement learning to explore the environment and come up with efficient actions to execute in complex task environments. Due to this exploration and joint optimization, PEAR significantly outperforms the imitation learning baselines, as stated in the Experiment Section 5 and Table 1. Moreover, PEAR mitigates non-stationarity by periodically re-populating subgoal transition dataset $D_g$ after every $p$ timesteps according to the goal achieving capability of the current lower primitive. This generates a natural curriculum of achievable subgoals for the lower primitive, thereby ameliorating non-stationarity. We hope that this elucidates the motivation behind how and why PEAR deals with non-stationarity. Please let us know and we would be happy to clarify any other doubts and concerns regarding the motivation.
>
> **2. "[Performance limitation from expert] Since the high-level policy is optimized only through BC or IRL using expert samples and does not participate in the environment exploration process, the performance of expert samples significantly limits the upper-performance limit of PEAR. Therefore, the authors need to analyze the impact of expert performance on the upper-performance limit of PEAR and conduct experiments with expert samples of varying performance."**
>
> Response:
>
> As clarified in the previous answer, PEAR employs reinforcement learning along with imitation learning regularization to explore the environment and devise efficient actions to execute in the complex task environments. Due to this, PEAR significantly outperforms various hierarchical and non-hierarchical baselines that may or may not employ imitation learning. Additionally, we perform ablations to analyze the effect of varying the number of expert demonstrations for each task (Please refer to Appendix A.4 Figure 9). Although it is subject to availability, we increase the number of expert data until there is no significant performance boost. We also conduct experiments where we vary the quality of demonstrations by adding in some g number dummy/garbage demonstrations with the N expert demonstrations, but since the algorithm is unable to learn from such garbage demonstrations, empirically the method performs as if it is being trained using (N-g) demonstrations, which is similar to the ablations analyzing the effect of varying the number of expert demonstrations for each task, and hence we omit the ablation results from the paper.
>
>
> **3. "[How to get $Q_{thresh}$] When performing adaptive relabeling, the authors require additional environment-specific Q values, which are not available in standard experimental settings and may limit the practicality of PEAR."**
>
> Response:
>
> The proposed approach does require setting $Q_{thresh}$ hyper-parameter but we empirically found that the performance is not unstable with varying hyperparameter value. Intuitively, this is a reasonable trade-off that renders the higher level subgoal generation a self-supervised process, thereby eliminating the need for expert guided annotations for segmenting subgoals. Additionally, please note that we empirically found that after we normalize $Q_{\pi^L}$ values of a trajectory, the $Q_{thresh}$ hyper-parameter value of $0$ works consistently well across all environments, which exhibits that setting $Q_{thresh}$ value does not cause significant overhead. We have added the hyper-parameter ablation analysis in Appendix A.4 Figure 10 for your consideration. Additionally, in real robotic experiments, we used the default $Q_{thresh}$ value of $0$ and PEAR was able to outperform the baselines in all experiments.

---

> > ### Author Response · Authors · 2023-11-17
> > **Author Response 2/2:**
> >
> > **4. "Why does the additional margin classification objective in low-level policy optimization prevent over-estimation?"**
> >
> > Response:
> >
> > As explained in Section 4.1, if the expert states are outside the training distribution, $Q_{\pi_L}$ might erroneously over-estimate the values on out-of-distribution states, which might result in poor subgoal selection. This surrogate objective prevents over-estimation of $Q_{\pi_{L}}$ by penalizing states that are out of the expert state distribution. Please note that the additional margin classification objective is only used with the higher level policy optimization and not with the lower level policy optimization objective. The ablation results demonstrating the importance of margin classification objective is shown in Figure 5.
> >
> > **5. "In equation (2), why does the low-level policy need the BC regularization objective? In my view, the goal of the low-level policy is to reach sub-goals more effectively, and this regularization is confusing."**
> >
> > Response:
> >
> > In complex tasks, specially in sparse reward scenarios, it is hard for the flat single level policy to achieve the goal, and using expert demonstrations to bootstrap learning using behavior cloning has been shown to improve performance [https://arxiv.org/abs/1709.10089]. Notably, we also found this to be the case in our empirical analysis (Please refer to Table 1). The higher level policy expert dataset provides subgoal targets and the lower level expert dataset provides primitive actions targets for behavior cloning training. Thus, the BC regularization objective allows the lower level to reach the goals more effectively by making use of expert primitive actions from the demonstration trajectories using behavior cloning (BC) or inverse reinforcement learning IRL) regularization.
> >
> > **6. "Why is the joint value function formulated as a summation of $J$ and $J_{BC}$?"**
> >
> > Response:
> >
> > The joint optimization objective for learning hierarchical policies is formulated as a sum of $J$ and $J_{BC}$. Here, $J$ denotes the off-policy reinforcement learning objective, and $J_{BC} denotes the behavior cloning or inverse reinforcement learning regularization objective. This joint optimization allows the agent to explore the environment and collect experience for the replay buffer, while leveraging expert dataset supervision using IRL or BC. Thus, this joint optimization objective comprising of RL and BC/IRL, combined with hierarchical training provide our method with the tools to solve complex long horizon tasks and outperform the baselines.
> >
> > **7. "The clarity of the figures in the experiments is poor, especially in Figure 5."**
> >
> > Response:
> >
> > We apologize for the figure size and clarity issues. We have now fixed the issues in the paper. We hope this improves the overall clarity.
> >
> > **8. "In section 3, the authors define the high-level reward, $r_{ex}$. However, the subsequent high-level policy optimization process does not appear to utilize $r_{ex}$." Please clarify the purpose of the high-level reward."**
> >
> > Response:
> >
> > The high level policy is trained using joint optimization objectives $J$ and $J_{BC}$, where $J$ denotes reinforcement learning term and $J_{BC}$ denotes imitation learning regularization. The higher level policy gets extrinsic reward $r_{ex}$ while exploring the environment, and this reward is used to train the higher policy using off-policy reinforcement learning. We use off-policy Soft Actor Critic algorithm in our experiments.
> >
> >
> > **9. "I recommend that the authors survey and include some of the most recent works in the related works section."**
> >
> > Response:
> >
> > We thank the reviewer with the survey suggestions. We have surveyed the suggested papers and improved the Related works section by adding their discussion.
> >
> > **10. "In the experimental setup, since PEAR requires training with expert samples (possibly high-performance expert samples), while other baselines like HAC do not depend on expert data, please clarify the fairness of the experimental comparisons."**
> >
> > Response:
> >
> > The reviewer is correct to point out that PEAR leverages expert demonstration samples whereas HAC does not. HAC deals with non-stationarity by relabeling transitions and assuming an optimal lower primitive. Whereas, PEAR deals with non-stationarity by periodic adaptive relabeling and IL regularization. Notably, expert demonstrations can not be directly leveraged in raw trajectory form to train the higher level, and therefore require data relabeling to segment the them into meaningful subgoals. Thus, although we agree that the comparisons with HAC baseline are not exactly fair, comparion with HAC demonstrate that PEAR can better ameliorate non-stationarity using a handful of expert demonstrations by generating expert subgoal dataset using adaptive relabeling.
> >
> > We hope that the response addresses the reviewer's concern. Please let us know, and we will be happy to address additional concerns if any.

---

> > > ### Author Response · Authors · 2023-11-20
> > > **Discussion:**
> > >
> > > Dear reviewer,
> > >
> > > We hope our response clarified your initial concerns/questions. We would be happy to provide further clarifications where necessary.

---

> > > ### Comment · Reviewer_uEcd · 2023-11-20
> > > **Further discussion on Q8**
> > >
> > > You define $r_{ex}$ and claim it has been used for high-level learning. Could you point out the exact objective function that using it? As  $r_{ex}$ is not revealed in Eq(1)-(8) in your paper.

---

> > > > ### Author Response · Authors · 2023-11-21
> > > > **Discussion:**
> > > >
> > > > Dear reviewer,
> > > >
> > > > We hope our response clarified your initial concerns/questions. We would be happy to provide further clarifications where necessary.

---

> > ### Comment · Reviewer_uEcd · 2023-11-20
> > **Further discussion on Q2**
> >
> > What I want to know is not whether adding some negative samples to the expert data has a huge impact on performance, as this is easy to go through the classifier. What I want to know is, if your expert policy is a RANDOM policy or MEDIUM expert (refer to D4RL datasets), how much does your performance change? Or, how much is your performance improvement over expert policy?

---

> ### Comment · Reviewer_uEcd · 2023-11-20
> **Further discussion on Q7**
>
> The clarity of the figures in the experiments is still poor in your revision. It is hard for the readers to read the lines and legends.

---

> > ### Author Response · Authors · 2023-11-20
> > **Author Response:**
> >
> > **"The clarity of the figures in the experiments is still poor in your revision. It is hard for the readers to read the lines and legends."**
> >
> > Response:
> >
> > We apologize for the figure size and clarity issues. We have now fixed the figures in the experiment section. We have also added an improved legend to enhance the figure clarity.

---

> > > ### Comment · Reviewer_uEcd · 2023-11-21
> > > **The figures still need more refinements**
> > >
> > > It seems there might be some misunderstandings regarding the figures' clarity. It would be beneficial to consult the figures in other published papers. The figure appear to be some screenshots, which present several issues: they are blurry, the resolutions are not optimal, and there are noticeable stretching. Also, the error bar's color doesn't match the corresponding mean value's solid line, leading to confusion. Furthermore, the tick labels are difficult to read, negatively impacting the paper's readability and overall quality.

---

> > > > ### Author Response · Authors · 2023-11-21
> > > > **Author response to "The figures still need more refinements"**
> > > >
> > > > We thank the reviewer for the helpful comments and suggestions. We have improved the clarity of the figures based on the reviewer's comments.

---

> ### Author Response · Authors · 2023-11-20
> **Author Response:**
>
> **"What I want to know is not whether adding some negative samples to the expert data has a huge impact on performance"**
>
> Response:
>
> We conducted separate experiments where we incrementally added bad/negative demonstrations to the expert demonstration dataset and compared its success rate performance. As expected, the performance degrades when the quality of number of bad demonstrations increases. However, since the approach also uses reinforcement learning along with the imitation learning objective, the performance drop is not significant and the policy is still able to show robustness to bad demonstration trajectories. We have added an ablation study in Appendix Figure 11 for maze navigation, pick and place, rope manipulation and franka kitchen environments (The experiments for bin and hollow environments are still running and we will upload the results soon).

---

> ### Author Response · Authors · 2023-11-20
> **Author Response:**
>
> **"You define $r_{ex}$ and claim it has been used for high-level learning. Could you point out the exact objective function that using it? As $r_{ex}$ is not revealed in Eq(1)-(8) in your paper."**
>
> Response:
>
> As briefly mentioned in Section 3 para 1 in the paper, the overall off-policy reinforcement learning objective is to maximize expected future discounted reward distribution: $ J = (1-\gamma)^{-1}\mathbb{E}_{s \sim d^{\pi}, a \sim \pi(a|s,g), g \sim G}\left[r(s_t,a_t,g)\right]$.
>
> For the high level policy, $r_{ex}$ will denote $r(s_t,a_t,g)$, which is the the extrinsic reward that the higher level policy receives from the environment. This represents the RL objective $J$ which is used in the joint optimization objectives in  Eqns 5-8. The reviewer is right to point out that we do not explicitly use the term $r_{ex}$ later in the paper which may cause confusion, but we used this term to clearly differentiate between the higher level extrinsic reward $r_{ex}$ (the reward that the higher level policy achieves from the environment), and the lower level intrinsic reward $r_{in}$ (the reward that the lower policy gets from the higher level policy) in the hierarchical setup. We hope that this clarifies the doubt. Please feel free to let us know and we would be happy to further clarify.

---

> ### Comment · Reviewer_uEcd · 2023-11-21
> **Further discussion on ""What I want to know is not whether adding some negative samples to the expert data has a huge impact on performance""**
>
> I am a bit confused by the response. I guess the authors ignore the "not" term in my question.
>
> I'd like to know the effectiveness of your method in two different cases.
> First, when using a RANDOM policy or a MEDIUM expert policy dataset from the D4RL datasets, how does your performance vary?
> Second, compared to the expert policy, how much improvement does your performance exhibit?
> Just adding some negative samples to the expert data is not expected, as this can easily be done with a classifier or low-weighted when RL training.

---

> > ### Author Response · Authors · 2023-11-21
> > **Further discussion Response:**
> >
> > We apologize for the confusion in previous replies. We provide the consolidated response to Q2 as follows:
> >
> > PEAR firstly generates higher level subgoal dataset using a handful of expert demonstrations, and then employs (i) reinforcement learning for reward based learning via exploration, and (ii) imitation learning regularization of higher policy using the previously generated higher level subgoal dataset (and lower level policy using primitive expert demonstration dataset). In our experiments, we consider multiple sparse reward environments which require complex long term decision making, where RL based approaches are unable to solve the tasks. This is also shown in Table 1, where single level or multiple level policies using vanilla RL fail to show good performance.
> >
> > As explained in Section 4, PEAR requires a small number of expert demonstrations to autonomously generate expert subgoal dataset and consequently leverage it using RL and imitation learning regularization. When the expert demonstrations are sub-optimal, we expirically found that the performance of PEAR degrades. However, even when the expert demonstrations are completely RANDOM, PEAR is still able to learn using reward based exploration using RL. We also trained an expert policy using imitation learning leveraging the expert demonstrations. However, PEAR was able to consistently outperform the baseline as shown below:
> >
> > | Method             | Maze            | Pick and Place  | Bin           | Hollow         | Rope           | Kitchen        |
> > |--------------------|-----------------|-----------------|---------------|----------------|----------------|----------------|
> > | PEAR-IRL           | 0.84 $\pm$ 0.04 | 0.92 $\pm$ 0.02 |0.79 $\pm$ 0.05|0.78 $\pm$ 0.27 |0.33 $\pm$ 0.04 |0.89 $\pm$ 0.06 |
> > | PEAR-BC            | 0.67 $\pm$ 0.07 | 0.48 $\pm$ 0.03 |0.38 $\pm$ 0.19|0.33 $\pm$ 0.03 |0.32 $\pm$ 0.04 |1.0 $\pm$ 0.0   |
> > | Expert using IL    | 0.              | 0.              | 0.            | 0.             | 0.15           | 0.             |

---

> > > ### Comment · Reviewer_uEcd · 2023-11-22
> > >
> > > Thank the authors for their rebuttal. I would raise my score.

---

> > > > ### Author Response · Authors · 2023-11-22
> > > > **Author Response:**
> > > >
> > > > We would like to thank the reviewer for being patient and for taking the time to review the paper and provide very helpful comments which definitely led to an improved paper draft and improved understanding of our approach.

---

### Official Review · Reviewer_XmJk · 2023-11-01

**Soundness:** 2 fair
**Presentation:** 2 fair
**Contribution:** 3 good
**Rating:** 5
**Confidence:** 3

**Summary:**

The paper proposes a combination of hierarchical reinforcement learning (HRL) and imitation learning (IL) for solving complex long-horizon tasks. A small number of expert demonstrations consisting of sequences of states is assumed to be available. The proposed hierarchy consists of a higher level deciding on subgoals and a lower level trying to achieve these subgoals. Target subgoals are extracted from an expert demonstration by plugging states from the trajectory into the low-level Q-function and using its value as a proxy for reachability. The last state which still has a Q-value above a threshold is selected as subgoal and the procedure, referred to as “primitive enabled adaptive relabeling”, is repeated starting from this state. This results in a sequence of expert subgoals. This relabeling procedure is repeated periodically to take the improved performance of the lower level into account. The authors furthermore provide a bound for the suboptimality of the learned policies. Experiments in simulated and real robotics environments show good performance of the proposed method.

**Strengths:**

The proposed method for extraction of expert subgoals from expert demonstrations is well motivated and intuitive. The integration of expert demonstrations into HRL frameworks is furthermore a promising research direction as demonstrated by the good empirical results.

The paper moreover provides a wealth of empirical results in simulated and real robotics environments. The conducted ablation studies add to an understanding of how to choose the hyperparameters of the method.

**Weaknesses:**

Unfortunately, some parts of the paper are difficult to understand or even seem contradictory. For example, in section three it is stated that no actions are part of the demonstrations:
> Notably, we only assume access to demonstration states and not actions.

However, in section 4.2 expert data $(s^f, a^f, s^f_\text{next})$ is used for imitation learning on the lower level. $a^f$ is explicitly used as an expert action here.

In Algorithm 1 in line 12 there are furthermore triplets $(s_j, s_w, s_k)$ added to the expert subgoal dataset $D_g^e$ where $k$ runs from the initial to the current goal index. Adding $s_k$ seems to be unnecessary as in section 4.2 only the first two entries in the triplet are actually used for imitation learning. It therefore seems to me that Algorithm 1 could be significantly simplified. In the second to last paragraph of section 4.1 there are furthermore triplets $(s_0^e, s_i^e, a_i)$ sampled from $D_g$ to train the lower level. This is again in contradiction to what was actually added to the dataset $D_g$.

In section 4.2 BC parameters $\zeta_L$ and $\zeta_H$ for the lower and higher parameters are introduced which are distinct from the parameters $\theta_L$ and $\theta_H$ of the low- and high-level policies. It is not clear what is meant by that. For example, the BC objective in equation (1) optimizes $\zeta_H$ but nothing in it actually depends on $\zeta_H$.

The theoretical analysis in section 4.3 is unfortunately really difficult to follow. For example, consider the two sentences:
> Let $\Pi^H_D$ and $\Pi^L_D$  be higher and lower level probability distributions which generate datasets $D_H$ and $D_L$ , and $\pi^H_D$ and $\pi^L_D$ are policies from datasets $D_H$ and $D_L$ . Although $D_H$ and $D_L$ may represent any datasets, in our discussion, we use them to represent higher and lower level expert demonstration datasets.

So $\Pi^H_D$ is a probability distribution which generates a dataset $D_H$ but then $\pi^H_D$ is a policy from $D_H$. So does $D_H$ consist of policies then? Apparently not because the second sentence says it is representing the expert dataset. So how can $\pi^H_D$ be from that dataset then? Is it some kind of empirical policy only defined on the data? This is completely unclear at this point.

There is also the issue of the distribution $\kappa$. It is not quite clear what it represents but I assume it is either the expert data or the distribution induced by the current policy. This would mean that the factor $\lVert \frac{d_c^{\pi*}}{\kappa} \rVert_\infty$ would almost certainly be infinite or at least ridiculously large which would render the bound completely vacuous.

There are several claims that PEAR mitigates the non-stationarity issue of HRL (in the introduction and in section 4.1). However, the transitions in the replay buffer of the SAC algorithm used on the higher level are getting outdated when the lower level changes. This is not addressed by PEAR so it seems questionable to me whether the algorithm really mitigates non-stationarity.

When it comes to the experiments it is great that many environments have been considered but it seems like the baselines are not suitable for these tasks. The rewards seem to be extremely sparse in the more complex environments which makes it very difficult to make any learning progress without demonstrations. I would therefore suggest to incorporate strong IL baselines into the experiments. It is also not quite clear how the hyperparameters for PEAR and the baselines have been tuned.

In Figure 5 there are no shaded regions in the second and third row. Does that mean that only one seed was used for these experiments? That might mean the results are too noisy to be able to interpret them properly.

The overall writing seems unfinished in some parts of the paper. For example, articles are frequently missing or verbs are singular when they should be plural or vice versa. These issues are already present in the first paragraph of the introduction and continue throughout the paper. The plots in figures 3 and 5 and some of the plots in the appendix are furthermore way too small to be legible when printed out.
The use of notation is unfortunately not consistent throughout the paper. For example, the threshold for the Q-function during relabeling has been introduced as $Q_\text{thresh}$ in section 4.1 but in section 5 in the paragraph “Ablative analysis” a $D_\text{thresh}$ makes an appearance. I would assume they refer to the same thing but it is not entirely clear. There are also smaller problems with the notation like $G$ vs $\mathcal{G}$ for the goal space (which seems to be identical to the state space as states and goals are being subtracted from each other in section 3 but this is not made explicit).

Figure 2 is difficult to interpret as the subgoals appear to be somewhat arbitrary. Perhaps a video would be better suited for demonstrating the improvement of the subgoals over training.

While there seems to be enough material for a publication, it is unfortunately not presented in a comprehensible way. Parts of the paper are contradictory and unclear and I therefore cannot recommend it for acceptance in its current form.

**Questions:**

* Why is the lower level referred to as a primitive?

---

> ### Author Response · Authors · 2023-11-17
> **Author Response Part 1/3:**
>
> We thank the reviewer for their detailed, constructive feedback. We address the reviewer's concerns as follows:
>
> **1. "For example, in section three it is stated that no actions are part of the demonstrations: Notably, we only assume access to demonstration states and not actions. However, in section 4.2, expert data $(s^f, a^f, s^f_{next})$ is used for imitation learning on the lower level. $a^f$ is explicitly used as an expert action here."**
>
> Response:
>
> When we train the higher level policies using IRL or BC, the higher level joint optimization procedures (Eqn 1 and Eqn 3) do not require expert actions $a^f$, which is what we initially wanted to convey through the statement. However, expert actions are indeed required when lower level policies are trained, using IRL or BC joint optimization. We understand that our previous statement may lead to confusion for the reader, and thank the reviewer for pointing this out. We have removed the statement and clarified in the paper draft wherever applicable (Section 3 and otherwise).
>
> **2. "In Algorithm 1 in line 12, there are furthermore triplets $(s_j,s_w,s_k)$ added to the expert subgoal dataset $D^e_g$ where runs from the initial to the current goal index. Adding $s_k$ seems to be unnecessary as in section 4.2, only the first two entries in the triplet are actually used for imitation learning. It therefore seems to me that Algorithm 1 could be significantly simplified."**
>
> Response:
>
> Considering the triplets $(s_j,s_w,s_k)$, the reviewer is absolutely right that $s_k$ is not used in the joint optimization objectives. Nevertheless, we had added the triplets to the dataset $D^e_g$ so that the dataset transitions are consistent with the replay buffer transition storage format used by Soft Actor critic for off policy reinforcement learning. We could remove the $s_k$ entry from the transitions, which simplifies the notation and improves space complexity, but honestly we are not sure how this would simplify Algorithm 1 significantly. Please feel free to let us know if we missed something and we would be happy to add the improvements.
>
> **3. "In the second to last paragraph of section 4.1 there are furthermore triplets $(s^e_0, s^e_i,a_i)$ sampled from $D_g$ to train the lower level. This is again in contradiction to what was actually added to the dataset $D_g$."**
>
> Response:
>
> In the second to last paragraph of section 4.1, in $Q_{\pi^{L}}(s^e_0,s^e_i,a_i)$, $s^e_0$ and $s^e_i$ are sampled from dataset $D_g$, whereas $a_i = \pi^{L}(s^e_{0},s^e_i)$, which is the action generated by the lower primitive policy. We have also shown this explicitly in Algorithm 1 when computing $Q_{\pi^{L}}(s^e_{init},s^e_i,a_i)$ where $a_i = \pi^{L}(s^e_{i-1},s^e_i)$.
>
> **4. "In section 4.2 BC parameters $\zeta_{L}$ and $\zeta_{H}$ for the lower and higher parameters are introduced which are distinct from the parameters $\theta_{L}$ and $\theta_{H}$ of the low- and high-level policies. It is not clear what is meant by that. For example, the BC objective in equation (1) optimizes $\zeta_{H}$ but nothing in it actually depends on $\zeta_{H}$."**
>
> Response:
>
> Initially, we had used $\zeta_{L}$ and $\zeta_{H}$ parameters to explicitly depict the policy parameters in the BC objective, but we understand now that this causes confusion, as correctly pointed out by the reviewer. Hence, we have removed the $\zeta$ variables from the draft and now $\theta$ are used to consistently depict the policy parameters.
>
> **5. "So $\Pi_{D}^{H}$ is a probability distribution which generates a dataset $D_H$ but $\pi_{D}^{H}$ then is a policy from $D_H$. So $D_H$ does consist of policies then? Apparently not because the second sentence says it is representing the expert dataset. So how can $\pi_{D}^{H}$ be from that dataset then? Is it some kind of empirical policy only defined on the data? This is completely unclear at this point."**
>
> Response:
>
> $\Pi_{D}^{H}$ represents some unknown distribution over policies from which we can sample a policy $\pi_{D}^{H}$. The expert dataset $D_H$ does not consist of policies; rather, it consists of state action transitions. Let us assume that these transitions are generated from an unknown empirical policy $\pi_{D_H}^{H}$. Thus, if we know $\pi_{D_H}^{H}$, we can use it to predict actions for the state transitions present in the dataset $D_H$. We hope that this clarifies the confusion.

---

> > ### Author Response · Authors · 2023-11-17
> > **Author Response Part 2/3:**
> >
> > **6. "There is also the issue of the distribution $\kappa$. It is not quite clear what it represents but I assume it is either the expert data or the distribution induced by the current policy. This would mean that the factor $\| \frac{d_c^{\pi^{*}}}{\kappa} \|_{\infty}$ would almost certainly be infinite or at least ridiculously large which would render the bound completely vacuous"**
> >
> > Response:
> >
> > $\kappa$ represents some unknown distribution over states from which we can sample a state $s$. In our sub-optimality bounds, we use it for importance sampling when we sample from arbitrary state distribution $\kappa$ over unknown distribution $d_{c}^{\pi^*}$. The factor $\| \frac{d_c^{\pi^{*}}}{\kappa} \|_{\infty}$ represents the infinite norm ratio between these distributions. We have kept the notations consistent with the theoretical analysis in the paper OPAL[https://arxiv.org/pdf/2010.13611.pdf]. The reviewer is encouraged to refer to this paper for further clarifications.
> >
> > **7. "There are several claims that PEAR mitigates the non-stationarity issue of HRL (in the introduction and in section 4.1). However, the transitions in the replay buffer of the SAC algorithm used on the higher level are getting outdated when the lower level changes. This is not addressed by PEAR so it seems questionable to me whether the algorithm really mitigates non-stationarity."**
> >
> > Response:
> >
> > We propose to mitigate non-stationarity by periodically re-populating subgoal transition dataset $D_g$ after every $p$ timesteps according to the goal achieving capability of the current lower primitive. Since the lower primitive continuously improves with training and gets better at achieving harder subgoals, $Q_{\pi^L}$ always picks reachable subgoals of appropriate difficulty, which generates a natural curriculum of subgoals for lower primitive. Intuitively, $D_g$ always contains achievable subgoals for the current lower primitive, thereby mitigating non-stationarity caused by outdated lower level transitions. We have also already explained this in detail in Section 4.1 (Primitive enabled adaptive relabeling) and Algorithm 2.
> >
> > **8. "When it comes to the experiments it is great that many environments have been considered but it seems like the baselines are not suitable for these tasks. The rewards seem to be extremely sparse in the more complex environments which makes it very difficult to make any learning progress without demonstrations. I would therefore suggest to incorporate strong IL baselines into the experiments. It is also not quite clear how the hyperparameters for PEAR and the baselines have been tuned."**
> >
> > Response:
> >
> > Sparse rewards are relevant for tasks such as the robotic tasks considered in this paper where it is hard to come across dense rewards. Since learning in sparse reward scenarios is extremely difficult, we chose to deal with such challenging scenarios for this work. Moreover, dense reward may sometimes cause bias in learning task when the the rewards are not optimal, and thus sparse reward scenarios lead to generalizable and effective policies. We would like to also point out that multiple hierarchical and non-hierarchical baselines which may/may not use expert denostrations have been considered in the paper. Relay Policy Learning (RPL) and Discriminator Actor Critic(DAC) are imitation learning based multi-level and single-level baselines that leverage expert demonstrations. We have also added behavior cloning baseline in success rate performance comparison results.
> >
> > We agree with the reviewer that it is hard to learn in the absence of expert demonstrations, but in complex scenarios, the Flat single level policies are not sufficient and therefore recent approaches employ hierarchical learning and leverage expert demonstrations. We hope that this clarifies the motivation of working in complex sparse reward scenarios and reasons for employing the beaslines in the paper.
> > For selecting the hyper-parameters, firstly we select $100$ randomly generated environments for training, testing and validation. Using these, we compute success rate plots for various hyper-parameter values for selecting the best hyper-parameter values (the ablation analysis are shown in Appendix A.4).

---

> > > ### Author Response · Authors · 2023-11-17
> > > **Author Response Part 3/3:**
> > >
> > > **9. "In Figure 5 there are no shaded regions in the second and third row. Does that mean that only one seed was used for these experiments? That might mean the results are too noisy to be able to interpret them properly."**
> > >
> > > Response:
> > >
> > > The results were averaged over 5 seeds and for success rate performance comparisons in all the environments. For the ablations analysis, we averaged over 3 seeds and found that the results were consistent across seeds, but unfortunately due to resource and time constraints we could not compute the results over 5 seeds for all the ablation experiments. We promise to add the comparisons over multiple seeds before submitting the final draft.
> > >
> > > **10." articles are frequently missing or verbs are singular when they should be plural or vice versa."**
> > >
> > > Response:
> > >
> > > Thank you for pointing out the errors. We have improved the writing in the recent draft wherever we could find the errors. We hope that this improves the overall clarity of the paper. Kindly let us know if you find any improvements and we will be happy to implement them.
> > >
> > > **11. "The plots in figures 3 and 5 and some of the plots in the appendix are furthermore way too small to be legible when printed out."**
> > >
> > > Response:
> > >
> > > We apologize for the figure size and clarity issues. We have now fixed the issues in the paper. We hope this improves the overall clarity.
> > >
> > > **12. "the threshold for the Q-function during relabeling has been introduced as $Q_{thresh}$ in section 4.1 but in section 5 in the paragraph "Ablative analysis" a $D_{thresh}$ makes an appearance. I would assume they refer to the same thing but it is not entirely clear. "**
> > >
> > > Response:
> > >
> > > $Q_{thresh}$ and $D_{thresh}$ indeed refer to the same threshold parameter. We apologize for the typo and we have made changes to the draft to keep the notation consistent throughout.
> > >
> > > **13. "There are also smaller problems with the notation like $G$ vs $\mathcal{G}$ for the goal space (which seems to be identical to the state space as states and goals are being subtracted from each other in section 3 but this is not made explicit)"**
> > >
> > > Response:
> > >
> > > We thank the reviewer for pointing this out. $\mathcal{G}$ was a typo in Section 3. We have removed the typo and the notation is now consistent throughout the paper.
> > >
> > > **14. "Figure 2 is difficult to interpret as the subgoals appear to be somewhat arbitrary. Perhaps a video would be better suited for demonstrating the improvement of the subgoals over training."**
> > >
> > > Response:
> > >
> > > We understand that the subgoals improvement depiction in Figure 2 might not seem very clear, especially since the figures are presented in 2D. However, we indeed found that the subgoals complexity consistently improves with training. We are working to compile videos to demonstrate better depiction of improving subgoal and will add links to the video in the final draft.
> > >
> > > **15. "Why is the lower level referred to as a primitive?"**
> > >
> > > Response:
> > >
> > > The lower level policy takes primitive actions on the environments and is referred to as lower primitive in some recent works. Hence, in order to be consistent with the recent cited works we call the lower level policy, the primitive policy.
> > >
> > > We hope that the response addresses the reviewer's concern. Please let us know, and we will be happy to address additional concerns if any.

---

> ### Author Response · Authors · 2023-11-21
> **Discussy:**
>
> Dear reviewer,
>
> We hope our response clarified your initial concerns/questions. We would be happy to provide further clarifications where necessary.

---

> ### Comment · Reviewer_XmJk · 2023-11-21
>
> Response to rebuttal
>
> I thank the authors for their detailed answers to my questions and concerns.
>
> 1. Ok, great that this point could be clarified.
>
> 2. As reviewer hXvK also remarked, Algorithm 1 is quite hard to understand with its four nested four loops. Introducing $w=i-1$ as an additional variable does not seem to help with that nor does adding $s_w$ to $D_g^e$. You are right, removing these parts would not be a significant simplification but it would be a start for making the algorithm more readable (together with the comments you added).
>
> 3. I mean the second term where it explicitly says $(s_0^e, s_i^e, a_i) \sim D_g$ where $a_i$ is used as a low-level action. This seems to be in contradiction to Algorithm 1 which adds triplets of states to $D_g$.
>
> 4. Thank you for clarifying the notation. This makes it easier to understand the algorithm.
>
> 5. Thank you for the explanation. The formulation in the paper is not in line with what you write here, however. The paper reads “Let $\Pi^H_D$ and $\Pi^L_D$  be higher and lower level probability distributions which generate datasets $D_H$ and $D_L$ , and $\pi^H_D$ and $\pi^L_D$ are policies from datasets $D_H$ and $D_L$ .”, i.e., it explicitly states that $\Pi^H_D$ generates a dataset $D_H$ and $\pi^H_D$ is a policy from $D_H$. This paragraph does not properly introduce these objects but rather mixes them up.
>
> 6. Notation (like $\kappa$) should be introduced properly, in my opinion, even if it is also used in other papers. In the OPAL paper $\kappa$ is assigned a clear role (the initial state distribution), in Theorem 1 in the submission, it just appears without an explanation. It is not clear to me, why $\kappa$ as an arbitrary distribution makes sense in this context (and why the importance sampling ratio should be bounded).
>
> 7. What is usually meant by “the non-stationarity issue of HRL” is the changing behavior of the lower level during training. As the proposed algorithm uses the off-policy method SAC on the higher level, it suffers from this non-stationarity issue. Repopulating the subgoal dataset $D_g$ does not mitigate this kind of non-stationarity as it is caused by the lower level, not the higher level. That is not to say that the contribution of how $D_g$ is repopulated is not valuable, but the claim that PEAR mitigates the non-stationarity issue of HRL seems misleading to me.
>
> 8. I agree that sparse rewards are highly relevant. I just wanted to point out that the baselines without IL components do not stand a chance with very sparse rewards. It is good, however, that an IL baseline was considered. Thank you for the explanation regarding the hyper parameter tuning.
>
> 9. Thank you for the explanation and for adding missing seeds. I agree with Reviewer uEcd that the figures are still not ideal. The color of the shaded regions does not match the color of the mean, for example.
>
> 10. to 14. Great, thank you for your effort. I will give some examples where the article is missing. First paragraph of the introduction “a large amount of environment interactions”, “the higher level policy predicts subgoals”. In the same paragraph this should be plural like this “HRL suffers”. There are many more such small issues in the paper. I know these are small things but they are distracting while reading the paper. I would encourage the authors to try to fix these.
>
> Overall, the proposed algorithm seems interesting to me and the experimental results look good. However, I still have concerns about the presentation of the paper that could not be resolved by the response of the authors. I would appreciate further clarficiations.

---

> > ### Author Response · Authors · 2023-11-21
> > **Author Response:**
> >
> > We thank the reviewer for the helpful comments and detailed suggestions. We address the reviewer's concerns as follows:
> >
> > 2. **"Algorithm 1 is quite hard to understand with its four nested four loops":**
> >
> > Response:
> >
> > We have improved the algorithm as per the helpful suggestions. We have kept $s_k$ to avoid confusion for now, but if the reviewer thinks this adds confusion, we will promptly remove the term from the algorithm pseudo-code.
> >
> > 3. **"This seems to be in contradiction to Algorithm 1 which adds triplets of states to $D_g$."**
> >
> > Response:
> >
> > Actually, $D_g$ is the lower level expert dataset and $D^e_g$ is the higher level expert subgoal dataset. In the lower level expert dataset $D_g$, the triplets are of the form (current state, subgoal, primitive action), where the lower level policy takes 'primitive action' in the 'current state' to reach the 'subgoal' predicted by the higher level. Whereas, in higher level expert dataset $D^e_g$, the triplets are of the form (current state, final goal, predicted subgoal), where the higher level policy takes 'predicted subgoal' action in the 'current state' to achieve the 'final goal'. We hope that this clarifies the confusion.
> >
> > 5. **"The formulation in the paper is not in line with what you write here, however. "**
> >
> > Response:
> >
> > We agree with the reviewer that the explanation could cause confusion. We apologize for the confusion and we have improved the explanation in Section 4.3 in line with the previous explanation.
> >
> > 6. **"Notation (like $\kappa$) should be introduced properly, in my opinion, even if it is also used in other papers."**"
> >
> > Response:
> >
> > We agree to the reviewer and hence we have added a small explanation in Section 4.3 explaining the introduction and the purpose of $\kappa$. To clarify, we can use $\kappa$ in the importance sampling ratio to avoid sampling from the unknown $d_{c}^{\pi^{*}}$. We hope that this clarifies the doubt.
> >
> > 8. **"but the claim that PEAR mitigates the non-stationarity issue of HRL seems misleading to me."**
> >
> > Response:
> >
> > We agree with the reviewer that since the proposed algorithm uses off-policy method SAC on the higher level, it suffers from non-stationarity issue. However, when we state that the method mitigates non-stationarity, we imply that due to periodic re-polulation of $D_g$, since $D_g$ contains improved subgoal according to the current lower level primitive, the imitation learning regularization term allows the higher level policy to better predict subgoals. In the absense of such periodic re-polulation, the higher level expert policy may contain outdated transitions due to changing behavior of lower policy. However, we agree with the reviewer's sentiment and are be open to add explicit explanation or even statement re-phrasing  to avoid any such confusion.
> >
> > 9. **"I agree with Reviewer uEcd that the figures are still not ideal. The color of the shaded regions does not match the color of the mean, for example."**
> >
> > Response:
> >
> > We thank the reviewer for the helpful comments and suggestions. We have improved the clarity of the figures based on the reviewer's comments.
> >
> > 10. **" I would encourage the authors to try to fix these."**
> >
> > Reponse:
> >
> > The helpful suggestions from the reviewer are highly appreciated. We have improved the errors and typos wherever the reviewer suggested and where we could find them.
> >
> > We hope our response clarified your initial concerns/questions. We would be happy to provide further clarifications where necessary.

---

> > > ### Comment · Reviewer_XmJk · 2023-11-22
> > >
> > > 2. Thank you, I think the readability of Algorithm has been improved significantly compared to the initial version.
> > >
> > > 3. The notation $D_g$ is clearly used for the higher level expert subgoal dataset. This is evident in line 17 of Algorithm 1 where $D_g^e$ is added to $D_g$. It is also clear from lines 10 and 11 of Algorithm 2 which state that the higher level is trained using $D_g$. It looks to me like the same notation is used for two different things which causes confusion. I would encourage the authors to fix this.
> > >
> > > 5. Thank you for fixing the explanation of what $\pi^H_D$ and $D_H$ etc. mean. That makes the theory part more readable.
> > >
> > > 6. Ok, thank you for explaining the role of $\kappa$. It is not immediately clear what is gained by using importance sampling here as the $\lambda_H$ still depends on the distribution of $\pi^*$ and the infity norm of the importance sampling ratio can be huge.
> > >
> > > 9. and 10. Thank you for improving the figures and fixing some of the typos.
> > >
> > > Now that I have a better understanding of what the notation in the theory part means, I see an issue with the argument. The derivation up to equation (11) just suggests doing IL under the assumption that the expert data is near optimal.
> > >
> > > Technically, equation (10) can be rearranged into equation (12). Nevertheless, equation (12) is actually trivial because $J(\pi^*) \geq J(\pi^H_{\theta_H})$ by definition of the optimal policy. The additional terms that are subtracted on the RHS of equation (12) are greater or equal to zero which yields equation (12) without any need for the derivation done up to equation (11).
> > >
> > > So the issue I see is this: The theoretical analysis you did just suggests doing IL when the data is good enough. Equation (12) that you take as a motivation to also do RL is unfortunately trivial, and upon closer inspection, simply suggests to do RL. If you can optimize $J(\pi^H_{\theta_H})$ with RL, then you don't need IL. The RHS of equation 12 is essentially just a sum of the RL objective, a constant term and the IL objective for the higher level. This sum is ultimately ad-hoc, however. For this reason I am not quite convinced by section 4.3.
> > >
> > > I think the proposed algorithm is motivated well on an intuitive level but section 4.3 does not seem to add much to this while appearing a bit misleading.

---

> > > > ### Comment · Reviewer_XmJk · 2023-11-22
> > > >
> > > > I have raised my score in response to the improvements to the paper and the additional information provided by the authors. I still have concerns about the presentation, in particular the notation, and the alignment of the theory section with the actual algorithm.

---

> > > > > ### Author Response · Authors · 2023-11-23
> > > > > **Author Response:**
> > > > >
> > > > > Dear reviewer,
> > > > >
> > > > > We hope our response clarified your concerns/questions. We would be happy to provide further clarifications where necessary. Since the deadline is very close, please let us know if possible.

---

> > > > ### Author Response · Authors · 2023-11-22
> > > > **Author Response:**
> > > >
> > > > 3. **"The notation $D_g$ is clearly used for the higher level expert subgoal dataset. This is evident in line 17 of Algorithm 1 where $D_g$ is added to $D_g$. It is also clear from lines 10 and 11 of Algorithm 2 which state that the higher level is trained using $D_g$. It looks to me like the same notation is used for two different things which causes confusion. I would encourage the authors to fix this."**
> > > >
> > > > Response:
> > > >
> > > > We thank you for pointing out the error and apologize for the previous erroneous response. We have fixed the notation in the paper to prevent confusion. $D_g$ represents higher level expert dataset and $D$ represents lower level expert dataset. In the margin classification objective, lower policy $\pi^L$ is used to predict the primitive action $a_i$, and the notation of the samples from $D_g$ is now consistent throughout the paper.
> > > >
> > > > 5. **"It is not immediately clear what is gained by using importance sampling here as the $\lambda_H$ still depends on the distribution of $\pi^{*}$"**
> > > >
> > > > Response:
> > > >
> > > > The reviewer is right to point out that $\lambda_H$ depends on the distribution of $\pi^*$, however, $\lambda_H$ term is generally set as hyper-parameter in the experiments. For example, as explained in last paragraph of Section 4, when we consider $d$ as Jensen-Shannon divergence, the imitation learning objective takes the form of inverse reinforce-
> > > > ment learning (IRL) objective, and the $\lambda_H$ term is the learning rate hyper-parameter, which we set by selecting some $N$ randomly generated environments for training, testing and validation in our empirical studies.

---

> > > > ### Author Response · Authors · 2023-11-22
> > > > **Author Response:**
> > > >
> > > > **"issue with alignment of theory section with actual algorithm"**
> > > >
> > > > Response:
> > > >
> > > > We thank your for your comments and would like to clarify to motivate the significance of the theoretical analysis.
> > > >
> > > > The reviewer mentioned: "The derivation up to equation (11) just suggests doing IL under the assumption that the expert data is near optimal". However, we would like to point out that Equations 10 and 11 bound the sub-optimalities of higher policy $\pi_{\theta_{H}}^{H}$ and lower primitive $\pi_{\theta_{L}}^{L}$ respectively. These sub-optimality bounds are significant since they bound the performance of the algorithm's objective function with respect to the optimal policy.
> > > >
> > > > We agree with the reviewer that trivially, $J(\pi^{*}) \geq J(\pi_{\theta_{H}}^{H})$.
> > > >
> > > > However, when Equation 10 is re-arranged to equation 12, this results in a lower bound of $J(\pi_{\theta_{H}}^{H})$, which in turn is the RHS of equation 12, which consists of three terms: the constant term, the RL objective term and the IL objective term. Thus, as explained in Section 4.3, the lower bounded RHS term can be perceived as a minorize maximize algorithm which intuitively means: the overall objective can be optimized by $(i)$ maximizing the objective $J(\pi_{\theta_{H}}^{H})$ via RL, and $(ii)$ minimizing the distance measure $d$ between $\pi_{D}^{H}$ and $\pi_{\theta_{H}}^{H}$.
> > > >
> > > > This formulation serves as a plug-and-play framework where we can plug in RL algorithm of choice for our off-policy RL objective $J(\pi_{\theta_{H}}^{H})$, and distance function of choice for distance measure $d$ for our imitation learning objective. This ultimately yields various joint optimization objectives. Moreover, our sub-optimality bound provides an insight into the significance of primitive enabled adaptive relabeling. As explained in Section 4.3, when we consider the constant term, since the lower primitive improves with training and is able to achieve harder subgoals, and since $D_g$ is re-populated using the improved lower primitive after every $p$ timesteps, $\pi_{D_g}$ continually gets closer to $\pi^{*}$,
> > > >
> > > > resulting in decrease in the value of $\phi_D$. This implies that the suboptimality bound in Equation 10 gets tighter, and consequently $J(\pi_{\theta_{H}}^{H})$ gets closer to optimal $J(\pi^{*})$ objective.
> > > >
> > > > Additionally, although since $J(\pi^{*}) \geq J(\pi_{\theta_{H}}^{H})$, it may seem that equation 12 RHS is ad-hoc, however, as provided in Appendix A.1, we explicitly derive this specific formulation of equation 12, because the three terms on the RHS serve specific purposes:
> > > >
> > > > 1. the RL term can be optimized using an off the shelf off-policy RL algorithm (e.g. SAC),
> > > > 2. the IL regularization term can be optimized using an off the shelf IL algorithm (e.g. BC or IRL), and
> > > > 3. the constant term provides an intuition into why periodic re-population using our novel primitive enabled adaptive relabeling tightens the RHS bound, as $D_g$ improves according to the goal achieving capability of the current lower primitive. Thus, this provides an intuition into why our approach PEAR generates a curriculum of achievable subgoals for the lower primitive.
> > > >
> > > > Please note that although the sum of RL and BC objective has been studied before to improve single level policy training [https://arxiv.org/abs/1709.10089], our theoretical analysis provides an insight into why it performs well (in both single-level and hierarchical settings), and how it can be tweaked to generate various joint optimization objectives using our plug and play framework. Thus, our explanation bridges the intuitive explanations with theoretical analysis.
> > > >
> > > > We hope that this explanation clearly advocates the significance of our theoretical analysis and motivates that our theory section indeed aligns with the actual algorithm. We hope that our rebuttal is able to clearly elucidate and justify the hard work done on this project, and we hope to continue working in this exciting research area to gain more insights and keep pushing the boundaries of the current state of the art. We would like to thank the reviewer for being patient and for taking the time to review the paper and provide very helpful comments which definitely led to an improved paper draft and improved understanding of our approach.

---

> > > > > ### Comment · Reviewer_XmJk · 2023-11-23
> > > > >
> > > > > Thank you for addressing points 3. and 5. I think they are clear now.
> > > > >
> > > > > About the theory: The crux here is the rearrangement of equation (10) into equation (12) which seems to involve a step that results in a trivial equation.
> > > > >
> > > > > Since $J(\pi) - J(\pi^L_{\theta_L}) \geq 0$ we have for the RHS for equation (10), $|J(\pi) - J(\pi^L_{\theta_L})| = J(\pi) - J(\pi^L_{\theta_L})$. Hence, rearranging equation (10) should result in an upper bound for $J(\pi)$ like this: $J(\pi) \leq J(\pi^L_{\theta_L}) + \lambda_L\phi_D + \lambda_L \cdot (\ldots)$ and not in a lower bound like in equation (12). Equation (12) still holds trivially because of $J(\pi) - J(\pi^L_{\theta_L}) \geq 0$ but it is not actually derived from equation (10) and is also not particularly well motivated because anything greater or equal to zero could be subtracted from $J(\pi^L_{\theta_L})$ on the RHS and equation (12) would still hold.
> > > > >
> > > > > I hope this clarifies my concerns.

---

> > > > > > ### Author Response · Authors · 2023-11-23
> > > > > > **Author Response:**
> > > > > >
> > > > > > Thank you for providing clarifications to the doubts. We understand the concern now. When we derived the equations, we considered the magnitude on LHS and therefore thought that we could reverse the inequality. However, we understand the issue now. We have improved upon the statements to better explain the theory section.
> > > > > >
> > > > > > However, since Equation 10 and equation 12 still hold, and both equations have a valid reasoning: Equation 10 provides a bound for sub-optimality of the approach, and Equation 12 can be used to explain RL with IL regularization, while also explaining the effect of hierarchical curriculum learning using periodic re-population of expert dataset, the theory section still explains the motivations.
> > > > > >
> > > > > > Please consider this paper for submission. We feel that even after the corrections, the approach provides ample contribution to the literature of hierarchical curriculum learning with leveraging expert demonstrations. We respectfully request you to re-consider your score for the paper, since the deadline is in a few minutes.

---

### Author Response · Authors · 2023-11-17
**General response to reviewers and AC, and summary of updates to manuscript:**

We thank the reviewers for their constructive feedback. We are encouraged that the reviewers find our approach novel (Reviewer uEcd), well motivated and intuitive (Reviewers XmJk, cZN6, hXvK). We are glad that reviewers find our empirical results in real and simulation environments adequate and impressive (Reviewers XmJk, hXvK), and our ablation studies extensive and convincing (Reviewer XmJk).

Additionally, we are pleased that the reviewer (XmJk) believes the integration of expert demonstrations into HRL framework is a promising research direction. We also also glad that the reviewers identified our theoretical contributions (Reviewers uEcd, cZN6 and hXvK) and consider our theoretical analysis to be sound and thorough. We propose primitive enabled adaptive relabeling (PEAR), a hierarchical reinforcement learning and imitation learning based approach that performs adaptive relabeling on a handful of expert demonstrations to solve complex long horizon tasks. We have thoroughly addressed the reviewer's concerns and have tried to incorporate all feedback.

1. We have tried our utmost to respond with reviewer's concerns and clarify the motivation and the general reviewers' doubts regarding the general proposed approach.

2. We have removed any minor typos, notation errors and have improved clarity of figures to improve the overall clarity of the paper.

3. We have compared our approach to the previous and contemporary works suggested by reviewers, and have clearly specified our novelty and contributions in detail.

4. We have also surveyed the papers kindly suggested by the reviewers and have accordingly added and improved the related works section of the paper.

We hope that this clarifies all of the reviewer's doubts and concerns. Please let us know, and we will be happy to address additional concerns if any.

---

### Meta-Review · Area_Chair_vU6s · 2023-12-06

**Metareview:**

The paper presents PEAR, a method to address non-stationarity in hierarchical reinforcement learning (HRL). The method generates subgoals by adaptively relabelling expert demonstrations and then combines reinforcement learning and imitation learning to solve complex long-horizon tasks. The paper presents a theoretical bound on the sub-optimality of the approach and presents empirical results on robotics environments. After an extensive discussion the reviewers remained concerned about a distinction from related work, clarity of writing, and alignment of the theory and algorithm. We appreciate the efforts of the authors in improving the submission based on the reviewers' suggestions and feedback.

**Justification For Why Not Higher Score:**

Despite extensive discussions with the reviewers and multiple revisions there still seem to be some concerns about the paper, specifically; distinction from related work, clarity of writing, alignment of the theory and algorithm.

**Justification For Why Not Lower Score:**

N/A

---

### Decision · Program_Chairs · 2024-01-16

Reject